



# Contrasting chemical environments in summertime for atmospheric ozone across major Chinese industrial regions: the effectiveness of emission control strategies

Zhenze Liu[1], Ruth M. Doherty[1], Oliver Wild[2], Michael Hollaway[2,a], and Fiona M. O'Connor[3]

[1]School of GeoSciences, The University of Edinburgh, UK
[2]Lancaster Environment Centre, Lancaster University, UK
[3]Met Office Hadley Centre, UK
[a]now at: Centre for Ecology & Hydrology, Lancaster Environment Centre, UK

**Correspondence:** Zhenze Liu (zhenze.liu@ed.ac.uk)

**Abstract.** The UKCA chemistry-climate model is used to quantify the differences in chemical environment for surface $O_3$ for six major industrial regions across China in summer 2016. We first enhance the UKCA gas-phase chemistry scheme by incorporating reactive VOC tracers that are necessary to represent urban and regional-scale $O_3$ photochemistry. We demonstrate that the model with the improved chemistry scheme captures the observed magnitudes and diurnal patterns of surface $O_3$ concentrations across these regions well. Simulated $O_3$ concentrations are highest in Beijing and Shijiazhuang on the North China Plain and in Chongqing, lower in Shanghai and Nanjing in the Yangtze River Delta, and lowest in Guangzhou in the Pearl River Delta despite the highest daytime $O_3$ production rates in Guangzhou. $NO_x$/VOC and $H_2O_2$/$HNO_3$ ratios indicate that $O_3$ production across all regions except Chongqing is VOC limited. We confirm this by constructing $O_3$ response surfaces for each region changing $NO_x$ and VOC emissions and further contrast the effectiveness of measures to reduce surface $O_3$ concentrations. In VOC limited regions, reducing $NO_x$ emissions by 20 % leads to a substantial $O_3$ increase (11 %) in Shanghai. We find that reductions in $NO_x$ emissions alone of more than 70 % are required to decrease $O_3$ concentrations across all regions. Reductions in VOC emissions alone of 20 % produce the largest decrease (- 11 %) in $O_3$ levels in Shanghai and Guangzhou and the smallest decrease (- 1 %) in Chongqing. These responses are substantially different from those currently found in highly populated regions in other parts of the world, likely due to higher $NO_x$ emission levels in these Chinese regions. Our work provides an assessment of the effectiveness of emission control strategies to mitigate surface $O_3$ pollution in these major industrial regions, and emphasizes that combined $NO_x$ and VOC emission controls play a pivotal role in effectively offsetting high $O_3$ levels. It also demonstrates new capabilities in capturing regional air pollution that will permit this model to be used for future studies of regional air quality-climate interactions.

## 1 Introduction

Surface ozone ($O_3$) has become the main cause of atmospheric pollution in the summertime in China since 2013 and is particularly severe in industrial areas of China such as the North China Plain (NCP), the Yangtze River Delta (YRD), the Pearl River Delta (PRD) and the Sichuan basin where precursor emissions are high (Li et al., 2019a). The 90th percentile of the




maximum daily 8-hour average (MDA8) $O_3$ concentration in 30 of 74 major cities of China exceeded the National Ambient Air Quality Standard (100 ppb) in summer, 2017 (Wang et al., 2017; Lu et al., 2018; Silver et al., 2018; Li et al., 2019b; Lu et al.,

2019a). During 2013-2017, the national Air Pollution Prevention and Control Action Plan has successfully reduced emissions of fine particulate matter ($PM_{2.5}$) and nitrogen oxides ($NO_x = NO + NO_2$) in China by 33 % and 21 %, respectively (Zheng et al., 2018). However, the reduction in $NO_x$ emissions has led to an increase in $O_3$ levels in polluted areas due to the non-linear chemistry of $O_3$ and reduced titration of $O_3$ by NO (Li et al., 2019a; Wang et al., 2019). Volatile organic compounds (VOCs) are also important $O_3$ precursors and emissions of these have increased across China over the same period, exacerbating $O_3$

pollution (Zheng et al., 2018). VOC emissions are believed to have decreased in megacity regions such as Beijing (Cheng et al., 2019), but the resulting $O_3$ decrease is likely to have been offset by the $O_3$ increase due to reduced $NO_x$ emissions. This poses a challenge to mitigate surface $O_3$ pollution. Therefore, the balance of emission controls on $NO_x$ and VOC is critical to decreasing $O_3$ levels in these regions.

O$_3$ is a secondary photochemical pollutant in the troposphere that can be produced rapidly through oxidation of its precursors

$NO_x$, VOCs and carbon monoxide (CO) in the presence of sunlight. Power plants, industry, residences, and transport are the main anthropogenic sources of $NO_x$ and VOC emissions (Monks et al., 2015; Li et al., 2018a). Isoprene is the principal biogenic VOC and is released from trees, plants and crops (Sindelarova et al., 2014). $O_3$ formation is mainly initiated through oxidation of VOC species by hydroxyl radicals (OH). The resulting organic peroxy radicals ($RO_2$) and hydroperoxyl radicals ($HO_2$) can convert NO to $NO_2$ without destroying $O_3$. $O_3$ is then created from the combination of $O(^3P)$ atoms, formed from photolysis

of the resulting $NO_2$, and oxygen ($O_2$) (Sillman, 1999; von Schneidemesser et al., 2015; Wang et al., 2017). Under low $NO_x$ conditions, $HO_2$ radicals may combine to produce hydrogen peroxide ($H_2O_2$). However, at high $NO_x$ concentrations, nitric acid ($HNO_3$) and organic nitrates ($RO_2NO_2$) are easily formed as $NO_2$ reacts with OH and $RO_2$. $HNO_3$ and $RO_2NO_2$ are the main sinks of radicals and are readily removed from the atmosphere by deposition. Therefore, increasing $NO_x$ concentrations increase $O_3$ production, but also accelerate the formation of $HNO_3$ and $RO_2NO_2$, leading to less efficient $O_3$ formation. In

addition, direct titration of $O_3$ by NO becomes increasingly important at higher levels of $NO_x$. There is hence a transition in the magnitude of $O_3$ production from low to high $NO_x$ conditions. This turnover is dependent on the local chemical environment, and in particular on the relative abundance of $NO_x$ and VOCs (Sillman, 1995; Kleinman et al., 1997; Thornton et al., 2002; Kleinman et al., 2005; Sillman and West, 2009).

A variety of $O_3$ sensitivity indicators have been proposed to characterise the $O_3$ response to changing precursor emissions.

The simplest of these are based on the concentration ratios of the precursors, $NO_x$/VOCs, or of their oxidation products, $H_2O_2$/$HNO_3$ (Sillman, 1995). $O_3$ concentrations increase with $NO_x$ emissions and are not sensitive to VOC emissions in a $NO_x$ limited regime when $NO_x$ concentrations are relatively low (Sillman et al., 1990). However, in a VOC limited regime, $O_3$ levels may increase with decreasing $NO_x$ emissions, which is common in urban areas with high $NO_x$ emissions, and this is reflected in high $NO_x$/VOC or low $H_2O_2$/$HNO_3$ ratios. Critical values of these indicators of $O_3$ sensitivity vary by region

and by season (Sillman, 1995; Liu et al., 2010; Xing et al., 2019). Most major industrial regions in China are believed to be VOC limited and rural areas are $NO_x$ limited or in a transition regime (Jin and Holloway, 2015; Wang et al., 2017). $O_3$ production efficiency (OPE) is another important metric to evaluate the impacts of $NO_x$ emissions on $O_3$ concentrations (Liu





et al., 1987; Kleinman et al., 2002). This is defined as the number of $O_3$ molecules produced per molecule of $NO_x$ oxidised. Low OPE values are typically associated with high $NO_x$ conditions and indicate that there is less $O_3$ produced from a given amount of $NO_x$. OPE values generally increase as $NO_x$ emissions decrease, reflecting greater $O_3$ production per molecule of $NO_x$ oxidised at lower $NO_x$ levels.

In this study, we develop new capabilities in a global scale model by incorporating higher VOC chemistry, allowing the model to represent the oxidation environment in major industrialised regions in China. We focus on the spatial and temporal variation of daytime $O_3$ in this study. We first evaluate the performance of this global chemistry-climate model in simulating regional $O_3$ across large industrialised regions. We use $O_3$ sensitivity indicators to compare and contrast the chemical oxidative environment across these different regions in China to identify emission control measures that would be most beneficial to reduce $O_3$ pollution levels. Using a global model novelly allows us to compare the impact of emission control measures in China with those in other major industrialised regions across the world. The value of this approach is that the same model set-up can be used to assess the impact of future emission and climate scenarios, studies of tropospheric and stratospheric $O_3$ influences and comparisons of $O_3$ in different parts of world.

The configuration of the model used in this study is described in section 2, along with its development and application to surface $O_3$ in China. We evaluate the model performance in reproducing the diurnal cycles of surface $O_3$ and $NO_2$ in section 3, and we investigate the $O_3$ chemical environment in China, including $O_3$ precursor concentrations and sensitivity ratios in section 4. We calculate the local $O_3$ production rates, $O_3$ loss rates, $NO_x$ loss rates and OPE in section 5. We then quantify the $O_3$ responses to changing $NO_x$ and VOC emissions in these regions and investigate the requirements of emission controls to reduce $O_3$ levels in each region in sections 6 and 7. To provide a global context we compare and contrast the effectiveness of emission control strategies with that in other parts of the world in section 7 and present our conclusions in section 8.

## 2 Materials and methods

### 2.1 Model description, development and application

The United Kingdom Chemistry and Aerosols (UKCA) model is a state-of-the-art chemistry and aerosol model that simulates atmospheric composition from the troposphere to the upper stratosphere. It is coupled to the Met Office Hadley Centre's Global Environment Model (HadGEM) family of climate models, all of which are based on the UK Unified Model (MetUM) (O'Connor et al., 2014). It is also the atmospheric composition component of the UK Earth System Model (UKESM) (Sellar et al., 2019). Version 10.6.1 of UKCA is used in this study, coupled with the Global Atmosphere 7.1 (GA7.1) configuration (Walters et al., 2019) of HadGEM3 (Hewitt et al., 2011). The spatial resolution is N96L85 with 1.875° by longitude and 1.25° by latitude, and there are 85 terrain-following hybrid height layers with a model top at 85 km. The model time step is 20 minutes for meteorology, and chemistry is calculated every hour. Wind speed and temperature are nudged with ERA-interim reanalyses from the European Centre for Medium-Range Weather Forecasts (ECMWF) every 6 hours (Dee et al., 2011). Sea surface temperature and sea ice fields are prescribed with the climatology mean of 1995-2004 (Reynolds et al., 2007).



The Stratosphere-Troposphere (Strat-Trop) gas-phase chemical scheme is used to simulate the inorganic odd oxygen ($O_x$), hydrogen ($HO_x$ = OH + $HO_2$) and $NO_x$ chemical cycles, oxidation of CO and VOCs, chlorine and bromine chemistry, and heterogeneous processes on aerosols (Archibald et al., 2020). The Global Model of Aerosol Processes (GLOMAP) aerosol scheme is used with a two-moment pseudo-modal aerosol dynamics approach to simulate sulfate, sea-salt, dust, black carbon and both primary and secondary organic aerosol (Mann et al., 2010). Interactive photolysis is represented with Fast-JX which
derives photolysis rates between 177 and 750 nm (Neu et al., 2007).

Global chemistry-climate models typically include simplified gas-phase chemistry schemes representing a limited number of species to mitigate high computational demands. Major long-lived VOC species are selected and more reactive VOC species are typically omitted from the chemistry scheme (Young et al., 2018). Eight discrete emitted VOC species (formaldehyde (HCHO), ethane ($C_2H_6$), propane ($C_3H_8$), acetaldehyde ($CH_3CHO$), acetone (($CH_3$)$_2$CO), methanol ($CH_3OH$), isoprene
($C_5H_8$) and monoterpene ($C_{10}H_{16}$) are simulated in the Strat-Trop chemistry scheme of UKCA. This selection is appropriate for simulating the global burden of $O_3$ but is less suitable for simulating $O_3$ concentrations in high emission areas. In industrial regions of China, large abundances of more reactive VOCs such as alkenes and aromatics make substantial contributions to $O_3$ production (Wu and Xie, 2017; Tan et al., 2019; Liu et al., 2020). To address this, we incorporate more reactive classes of VOC including alkenes, higher alkanes and aromatics, represented by propene ($C_3H_6$), butane ($C_4H_{10}$) and toluene ($C_7H_8$)
respectively in the chemistry scheme (Atkinson et al., 2006; Folberth et al., 2006). This permits a more realistic simulation of photochemically active environments, and allows rapid $O_3$ formation in high VOC emission regions to be captured. The improved chemistry scheme includes 101 species, 244 bimolecular reactions, 26 uni- and termolecular reactions, 70 photolytic reactions, 5 heterogeneous reactions and 3 aqueous phase reactions for the sulfur cycle.

We perform model simulations for 2016 and focus our results on summer (June-July-August, JJA). We spin up the model
for 4 months and then simulate the full year nudged with ERA-interim reanalysis data for 2016. The new capabilities of the model allow us to investigate regional $O_3$ chemical environment in industrial regions of China in the model. The relatively coarse resolution of the model may lead to biases in surface $O_3$ associated with numerical diffusion (Wild and Prather, 2006; Stock et al., 2014; Fenech et al., 2018; Mertens et al., 2020), but we note that the lifetime of $O_3$ makes it a regional-scale pollutant except very close to high emission sources (Valari and Menut, 2008; Hodnebrog et al., 2011; Biggart et al., 2020).
This study demonstrates the first application of this improved chemistry scheme to high emission regions worldwide, and lays the foundation for more detailed studies of the interactions between air quality and climate in a global chemistry-climate model under future scenarios.

## 2.2 Emissions

The anthropogenic emission inventory of Hemispheric Transport of Air Pollution (HTAP) for 2010 is used for the globe
outside China (Janssens-Maenhout et al., 2015). The Multi-resolution Emission Inventory for China (MEIC) is used to provide emissions over China for 2013 (Li et al., 2017). We apply independent diurnal and vertical profiles to each emission sector (industry, power plants, transport and residences) according to European Monitoring and Evaluation Programme (EMEP) emissions (Bieser et al., 2011; Mailler et al., 2013). Biogenic VOC (BVOC) emissions are calculated interactively through



the Joint UK Land Environmental Simulator (JULES) land-surface scheme in UKCA (Pacifico et al., 2011). The Global
Fire Emissions Database (GFED4) are used for biomass burning emissions (van der Werf et al., 2010). Other aspects of the
emissions used are as described in Archibald et al. (2020).

Given the rapid changes in anthropogenic emissions across China, we adjust $NO_x$, VOCs, CO, sulphur dioxide ($SO_2$), black
carbon (BC) and organic carbon (OC) emissions in MEIC from 2013 to 2016 by applying national and urban emission scaling
factors. $NO_x$ emissions decreased by 18.8 % and VOC emissions increased slightly by 1.1 % between 2013 and 2016 across
China (Zheng et al., 2018). $NO_x$ and VOC emissions are estimated to have decreased by 24.2 % and 12.8 % respectively in
Beijing and surrounding areas between 2013 and 2016 (Cheng et al., 2019). We apply the Beijing scaling factors to major
industrialised regions, reflecting tighter emission controls in these developed urban regions, and use national scaling factors
across the rest of the country.

## 2.3 Selected regions and observations

We focus on six heavily populated regions with high emissions within the major industrialised regions in China. These include
Beijing and Shijiazhuang on the North China Plain (32 – 40 °N, 114 – 121°E), Shanghai and Nanjing in the Yangtze River
Delta (28 – 33 °N, 118 – 123 °E), Guangzhou in the Pearl River Delta (21 – 25 °N, 111 – 115 °E) and Chongqing in the Sichuan
Basin (28 – 32 °N, 103 – 108 °E), see Fig. 1. Anthropogenic $NO_x$ and VOC emissions are high in these regions (Fig. 2) due
to rapid industrialisation, urbanization and socio-economic development. Model grid cells that include observation stations
located in each of the urban and rural regions are selected to be representative of these regions, see Table 1. For comparison
with observations, we calculate a grid-weighted mean according to the number of measurement sites in each model grid cell
for the region.

We use observed hourly concentrations of air pollutants including $O_3$ and $NO_2$ from the surface monitoring networks of
China, obtained from the public website https://quotsoft.net/air/ which mirrors data from the Chinese National Environmental
Monitoring Centre (CNEMC) http://www.cnemc.cn/. There are 1,670 national measurement stations that started operating in
2013.





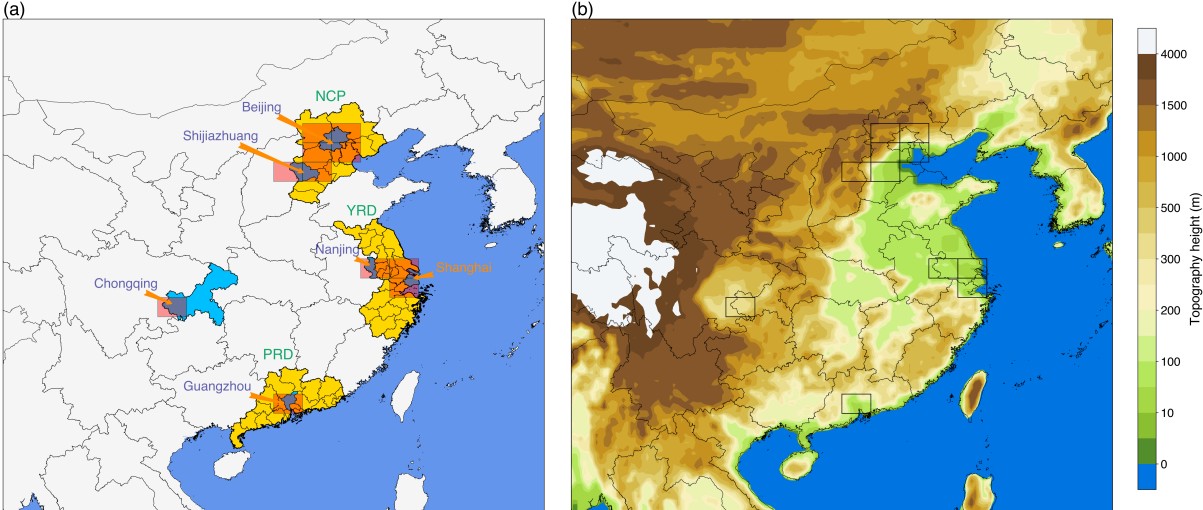

**Figure 1.** Map of China showing **(a)** the key provinces (yellow) representing the NCP, the YRD and the PRD and locations of the six regions (blue) – Beijing, Shijiazhuang, Shanghai, Nanjing, Guangzhou and Chongqing, and UKCA model grid cells co-located with these regions (red) **(b)** elevations across the whole of China.

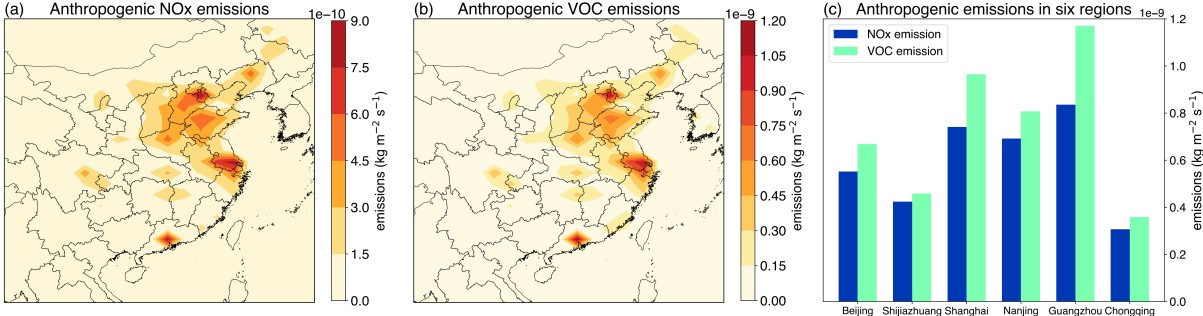

**Figure 2.** Spatial distributions of anthropogenic $NO_x$ and VOC emissions $(\mathrm{kg\,m^{-2}\,s^{-1}})$ across China **(a, b)** and grid-weighted averaged emissions for the six regions within the four major industrialised regions **(c)** in JJA, 2016.



**Table 1.** Number of measurement sites and grid cells in the six industrial regions

| Region | No. of measurement sites | No. of grid cells |
|---|---|---|
| Beijing | 46 | 4 |
| Shijiazhuang | 28 | 2 |
| Shanghai | 58 | 2 |
| Nanjing | 45 | 1 |
| Guangzhou | 45 | 1 |
| Chongqing | 25 | 1 |

## 3   Model evaluation of surface $O_3$ and $NO_2$

We evaluate the diurnal variation in simulated surface $O_3$ and $NO_2$ concentrations against summertime observations for JJA, 2016 for the six industrialised regions (Fig. 3, 4). In general, the diurnal variation of observed $O_3$ is matched relatively well and

the correlation coefficients are relatively high, see Table 2. Mean concentrations for $O_3$ and $NO_2$ over the lowest three model layers (from the surface up to 130 m) are also compared with observations. In the daytime, differences between the surface and three lowest layers are small due to efficient mixing in the planetary boundary layer (PBL). The height of the nocturnal PBL is typically underestimated in the model leading to overestimated $NO_x$ concentrations and hence underestimated $O_3$ concentrations at nighttime due to excessive $O_3$ titration by NO (André et al., 1978; Petersen et al., 2019; Zhao et al., 2019).

Fig. 3a shows a large difference in nighttime $O_3$ concentrations across the three layers, reflecting stable conditions that allow $NO_x$ to accumulate at the surface. Simulated surface $O_3$ concentrations therefore tend to be underestimated at nighttime. In contrast, the peaks in daytime $O_3$ concentrations are captured relatively well, reflecting efficient $O_3$ production in the high VOC environment.

Daily mean $O_3$ concentrations for Beijing, Shijiazhuang, Shanghai and Guangzhou are reproduced well with small biases

(~10 %; see Table 2). Simulated daily mean $O_3$ concentrations are highest (> 40 ppb) for Beijing, Shijiazhuang and Chongqing, lower in Shanghai and Nanjing (< 40 ppb), and lowest for Guangzhou (~30 ppb). Although daily mean $O_3$ concentrations are captured relatively well, as seen in Fig. 3a and 4a, daytime maximum $O_3$ concentrations are overestimated, associated with underestimated $NO_2$ concentrations. This overestimation is largest in Shijiazhuang where the underestimation of daytime $NO_2$ concentrations is larger than other regions. We find that there is a systematic bias in Chongqing where simulated $O_3$

levels are higher than observations. Chongqing is a mountainous inland region with complex topography that cannot be fully resolved, and surface $O_3$ here is thus representative of higher surface altitudes leading to a systematic bias high compared with observations (Su et al., 2018), and a corresponding bias low for $NO_2$ concentrations. In addition, simulated $O_3$ increases from biogenic emissions in the Sichuan basin are much larger in summertime than other regions (Lu et al., 2019b), and uncertainty in these emissions may contribute to the biases.

The diurnal patterns in $NO_2$ concentrations can also be captured as reflected by high levels at nighttime and low levels in the daytime for all regions. Daytime $NO_2$ concentrations can be reproduced relatively well, with a small underestimation. While





this may reflect underestimated $NO_x$ emissions, it is more likely to arise from the effects of dilution on $NO_x$. High emissions in these regions are diluted over a large grid cell, resulting in lower $NO_2$ concentrations in the daytime. This is offset by high $NO_2$ concentrations in the PBL at nighttime as discussed above. The diurnal variation of $NO_2$ concentrations is hence stronger

in the simulations than the observations (Fig. 4a).

Fig. 3 and 4 also show the frequency distribution of observed and modelled hourly $O_3$ and $NO_2$ concentrations. The simulated peaks in the distributions of $O_3$ and $NO_2$ are underestimated compared to observations for all six regions, reflecting the larger diurnal variation in the simulations. The diurnal variation is more closely simulated for $O_3$ concentrations (correlation coefficient r > 0.7) than for $NO_2$ concentrations. The Chongqing region has the closest correlation with observations (r = 0.83),

which provides evidence that the overestimation of $O_3$ is systematic as suggested earlier. Overall, the magnitudes (see Table 2) and diurnal patterns (see Fig. 3 and 4) of both species can be simulated reasonably well, with differences between industrial regions clearly captured.

**Table 2.** Comparison of modelled and observed daily mean surface $O_3$ and $NO_2$ concentrations for the six industrial regions in JJA, 2016, China.

| Region | Obs. (ppb) | Sim. (ppb) | Bias ppb/% | RMSE (ppb) | Correlation r |
|---|---|---|---|---|---|
| $O_3$ | | | | | |
| Beijing | 47.7±22.1 | 43.4±27.7 | -4.4 (9.1%) | 8.1 | 0.77 |
| Shijiazhuang | 42.9±18.4 | 47.6±28.7 | 4.7 (10.9%) | 11.6 | 0.78 |
| Shanghai | 38.3±17.5 | 34.4±28.8 | -3.9 (10.2%) | 12.7 | 0.77 |
| Nanjing | 42.6±18.9 | 35.9±24.9 | -6.8 (15.8%) | 9.8 | 0.71 |
| Guangzhou | 29.8±18.3 | 28.0±25.9 | -1.8 (-6.1%) | 9.4 | 0.81 |
| Chongqing | 38.1±19.2 | 56.0±31.3 | 18.0 (47.2%) | 22.3 | 0.83 |
| $NO_2$ | | | | | |
| Beijing | 17.8±3.7 | 20.7±8.2 | 2.9 (16.2%) | 5.8 | 0.69 |
| Shijiazhuang | 18.1±4.7 | 16.7±8.3 | -1.4 (7.7%) | 4.3 | 0.76 |
| Shanghai | 16.3±2.3 | 26.1±8.7 | 9.8 (60.0%) | 12.1 | 0.50 |
| Nanjing | 17.2±3.6 | 21.3±9.0 | 4.1 (23.7%) | 7.8 | 0.49 |
| Guangzhou | 16.1±3.0 | 19.9±9.3 | 3.8 (23.7%) | 8.4 | 0.55 |
| Chongqing | 17.4±3.8 | 10.9±6.3 | -6.4 (37.1%) | 8.0 | 0.43 |



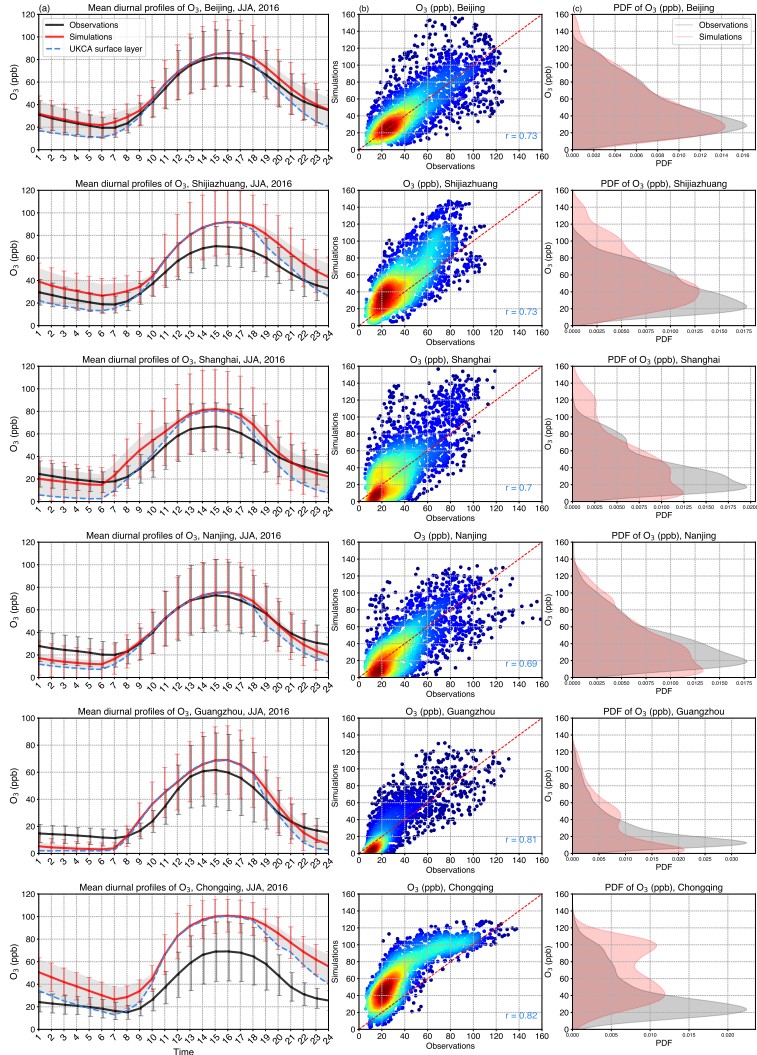

**Figure 3.** Comparison of observed and modelled O$_3$ concentrations for the six industrialised regions in JJA, 2016, China. **(a)** Mean diurnal cycles of observed and modelled O$_3$ concentrations (ppb). The shaded area represents the spread across the lowest three model layers. Error bars denote one standard deviation of hourly O$_3$ concentrations across all days **(b)** Observed and modelled hourly O$_3$ concentrations (ppb; three lowest model layers) and correlation coefficient values r **(c)** PDF of O$_3$ concentrations (ppb) for modelled and observed hourly values.



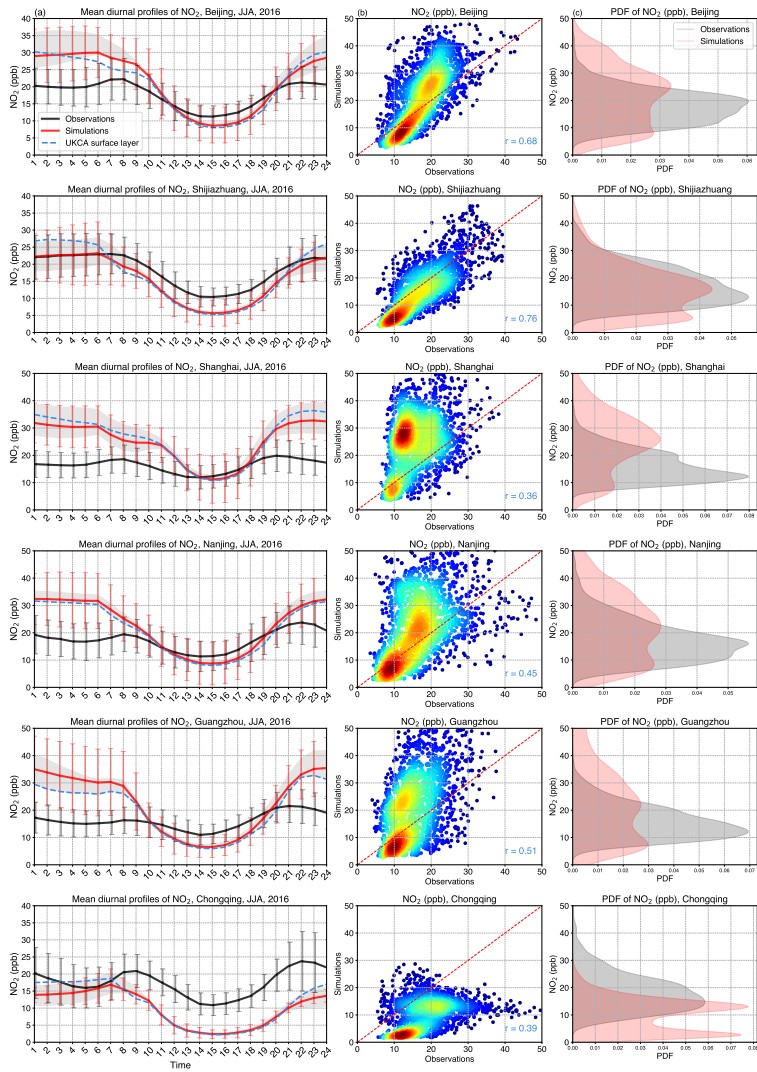

**Figure 4.** Comparison of observed and modelled $NO_2$ concentrations for the six industrialised regions in JJA, 2016, China. **(a)** Mean diurnal cycles of observed and modelled $NO_2$ concentrations (ppb). The shaded area represents the spread across the lowest three model layers. Error bars denote one standard deviation of hourly $NO_2$ concentrations across all days. **(b)** Observed and modelled hourly $NO_2$ concentrations (ppb; three lowest model layers) and correlation coefficient values r **(c)** PDF of $NO_2$ concentrations (ppb) for modelled and observed hourly values.

## 4 Differences in chemical environment

Spatial distributions of modelled daytime concentrations of $O_3$, $NO_x$, VOCs and $O_3$ sensitivity ratios ($NO_x$/VOCs and $H_2O_2$/$HNO_3$) are shown in Fig. 5 to illustrate the differences in chemical environment for the six regions. We focus on daytime hours with the highest $O_3$ concentrations using the MDA8 metric and consider this same time period for other species, which we refer to





hereafter as daytime concentrations. For the sensitivity ratio $NO_x$/VOCs we consider the sum of anthropogenic and biogenic daytime VOC concentrations.

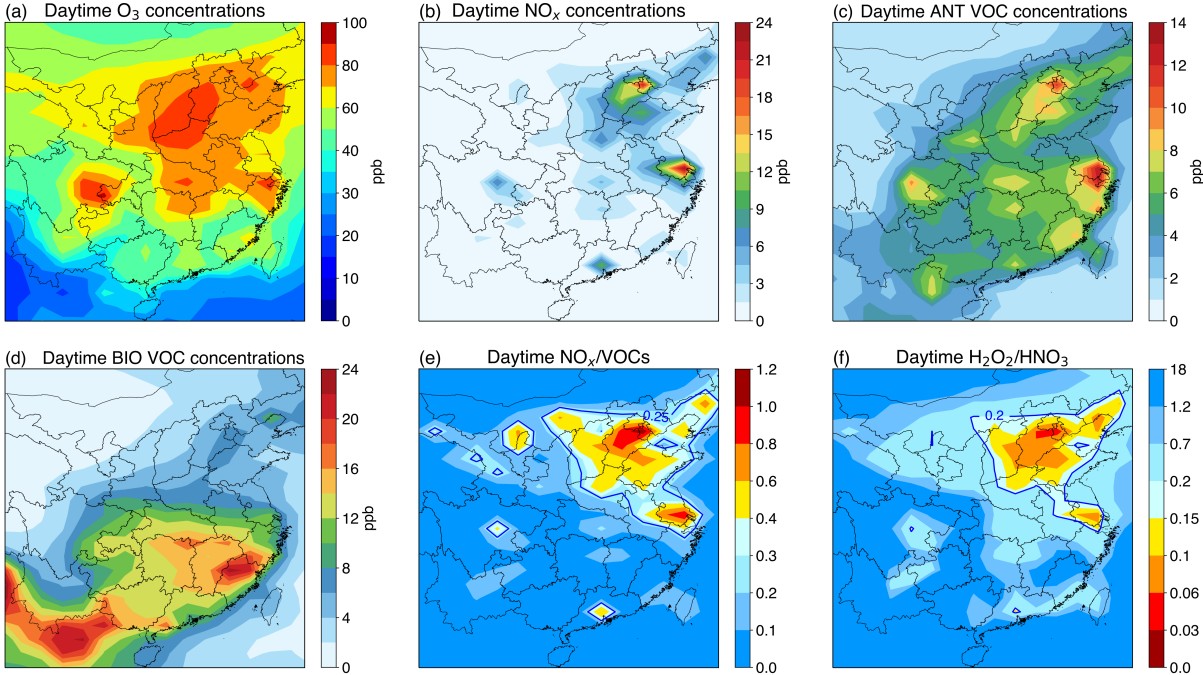

**Figure 5.** Spatial distributions of simulated surface daytime $O_3$, $NO_x$, anthropogenic VOCs, biogenic VOCs (ppb) **(a, b, c, d)** and two $O_3$ sensitivity ratios – $NO_x$/VOCs and $H_2O_2$/$HNO_3$ **(e, f)** in JJA, 2016, China.

Figure 5a shows high daytime $O_3$ levels (> 80 ppb) across northern China, eastern China and the Sichuan basin in JJA, 2016. $O_3$ levels in the PRD (~40 ppb) are much lower despite high emissions likely due to transport of clean air from the South China Sea associated with the East Asian summer monsoon (Zhao et al., 2010; Li et al., 2018b). Areas with high anthropogenic $NO_x$ and VOC concentrations generally coincide with high emission regions (Fig. 2, Fig. 5b, 5c). High daytime $NO_x$ concentrations (> 12 ppb) are simulated in Beijing and Shijiazhuang, Shanghai and Nanjing. Chongqing has the lowest $NO_x$ concentrations of 3–6 ppb due to relatively low $NO_x$ emissions. High anthropogenic daytime VOCs concentrations are simulated across the main industrial regions, in particular in Shanghai with the highest levels (> 12 ppb; Fig 5c).

The distribution of biogenic VOC concentrations differs from that of anthropogenic VOCs (Fig. 5c, 5d). There is a strong latitudinal gradient, reflecting differences in climate and the spatial distribution of vegetation (Li et al., 2013). The highest biogenic VOC levels (> 16 ppb) are simulated in south-eastern China where deciduous and mixed broadleaf trees are the main source of biogenic VOCs. The YRD, the PRD and the Sichuan basin have higher biogenic VOC concentrations than the NCP. Chongqing has the highest biogenic VOC levels (> 12 ppb) of the regions considered here. However, highest biogenic VOC levels are found south of China in Laos, Vietnam and Cambodia.





High $NO_x$/VOC ratios and low $H_2O_2$/$HNO_3$ ratios typically indicate VOC limited $O_3$ production. The transition between VOC and $NO_x$ limited regimes is typically about 0.25 for the $NO_x$/VOC ratio and about 0.2 for the $H_2O_2$/$HNO_3$ ratio (Liu et al., 2010; Xing et al., 2019). From these two thresholds for the $O_3$ sensitivity ratios, it can be seen that VOC limited regions cover most areas of the NCP, parts of the YRD including Shanghai and Nanjing, and Guangzhou in the PRD (Fig. 5e, 5f). All six regions except Chongqing have $NO_x$/VOCs ratios $\geq$ 0.6 and $H_2O_2$/$HNO_3$ ratios $\leq$ 0.18 (Table 3). This suggests that these regions have a chemical environment that is strongly VOC limited. In addition, VOC limited regimes shown by both indicators are quite similar, showing that these two $O_3$ sensitivity ratios may be useful to directly diagnose different $O_3$ sensitivity regimes in China. Regions with high $NO_x$/VOC ratios and low $H_2O_2$/$HNO_3$ ratios typically occur where $NO_x$ concentrations are high. Overall, these transition values delineate the different $O_3$ sensitivity regions across China well, showing VOC limited regimes in the major industrial regions with high emissions. However, we note that these $O_3$ sensitivity ratios only provide an estimate of the chemical environment, and further, more detailed investigation of $O_3$ responses to emission changes are required.

**Table 3.** Simulated surface daytime concentrations of species, radicals, $O_3$ sensitivity ratios and the photolysis rate $j(O^1D)$ for the six industrial regions in JJA, 2016, China.

| Region | Beijing | Shijiazhuang | Shanghai | Nanjing | Guangzhou | Chongqing |
|---|---|---|---|---|---|---|
| Species (ppb) | | | | | | |
| $O_3$ | 78.0 | 83.5 | 70.1 | 66.8 | 60.2 | 93.8 |
| $NO_x$ | 12.8 | 8.7 | 19.2 | 12.9 | 10.7 | 3.8 |
| VOC (ANT) | 8.7 | 7.0 | 12.7 | 7.6 | 7.5 | 7.7 |
| VOC (BIO) | 5.5 | 4.3 | 10.6 | 9.2 | 10.2 | 13.5 |
| VOC (Total) | 14.3 | 11.3 | 23.3 | 16.9 | 17.7 | 21.3 |
| Sensitivity ratios | | | | | | |
| $NO_x$/VOCs | 0.79 | 0.73 | 0.89 | 0.78 | 0.60 | 0.18 |
| $H_2O_2$/$HNO_3$ | 0.18 | 0.08 | 0.10 | 0.11 | 0.09 | 0.29 |
| Radicals | | | | | | |
| OH / $10^6$ cm$^{-3}$ | 7.8 | 10.3 | 8.4 | 9.8 | 13.0 | 16.6 |
| HO$_2$ / $10^8$ cm$^{-3}$ | 2.6 | 2.9 | 2.3 | 2.2 | 2.2 | 7.4 |
| RO$_2$ / $10^8$ cm$^{-3}$ | 1.0 | 0.9 | 0.8 | 0.8 | 0.9 | 2.5 |
| Photolysis rate | | | | | | |
| $j(O^1D)$/ $10^{-5}$ s$^{-1}$ | 2.3 | 2.6 | 2.3 | 2.5 | 3.1 | 3.4 |

## 5 Differences in local $O_3$ production rates

In this section, we calculate the daytime production rates for surface $O_3$ to investigate how the local $O_3$ production compares across the six regions. We define the net $O_3$ production rate (ppb/h) as the gross rate of production P($O_3$) from the reactions





$HO_2 + NO$ and $RO_2 + NO$ minus the gross rate of loss $L(O_3)$ from the reactions $O(^1D) + H_2O$, $O_3 + OH$, $O_3 + HO_2$ and $O_3 +$ VOCs. We assume that the pathways above represent the net $O_3$ production rate under $O_3$ photochemical steady state between NO and $NO_2$, and are shown as follows:

$$Net\ P(O_3) = P(O_3) - L(O_3) =$$
$$k_1[HO_2][NO] + k_2[RO_2][NO] - (k_3[O(^1D)][H_2O] + k_4[O_3][OH] + k_5[O_3][HO_2] + k_6[O_3][VOCs]) \quad (1)$$

where $k_i$ represents the rate coefficient of reaction i.

The loss of $NO_x$, $L(NO_x)$, is principally determined by the reactions $OH + NO_2$ and $RO_2 + NO_2$ which produce $HNO_3$ and $RO_2NO_2$, respectively. OPE is then defined as the number of $O_3$ molecules produced per molecule of $NO_x$ consumed (Liu et al., 1987).

$$OPE = \frac{P(O_3)}{L(NO_x)} \quad (2)$$

As shown in Fig. 6, local $O_3$ production varies across the six regions with $O_3$ net production rates ranging from 4-10 ppb/h. Simulated daytime net $O_3$ production rates are highest (> 8 ppb/h) in Shanghai and Guangzhou mainly due to high precursor emissions, and this is reflected by higher $O_3$ concentrations in Shanghai than in nearby Nanjing. While $O_3$ production is high in Guangzhou, the $O_3$ concentrations are much lower than in other regions, indicating that meteorological impacts in this coastal region are important to transport $O_3$ produced locally. $O_3$ net production in Beijing and Shijiazhuang is similar to that in Nanjing (~5 ppb/h). $O_3$ production in Chongqing is also high, reflecting high radical concentrations (see Table 3) that promote $O_3$ production despite lower precursor emissions. High photolysis rates $j(O(^1D))$ in Chongqing and Guangzhou contribute to high concentrations of OH radicals (Table 3). $O_3$ destruction rates are fairly similar (< 4 ppb/h) across these regions, but are higher in Chongqing, offsetting its high $O_3$ production rates.

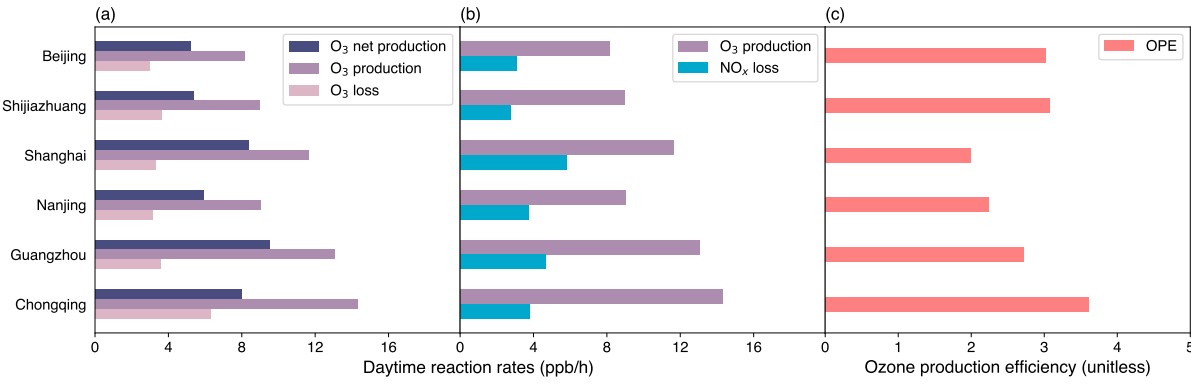

**Figure 6.** Simulated surface daytime **(a)** net $O_3$ production rates, gross $O_3$ production rates and gross $O_3$ loss rates (ppb/h) **(b)** gross $O_3$ production rates and $NO_x$ loss rates (ppb/h) **(c)** OPE (unitless) for the six industrial regions in JJA, 2016, China.





The simulated $NO_x$ loss rates (Fig. 6b) show the highest removal of $NO_x$ in Shanghai, where $NO_x$ concentrations are also highest. This influences OPE, which is strongly dependent on $NO_x$ loss, and leads to the lowest OPE in Shanghai and highest in Chongqing (Fig. 6c). The low OPE in Shanghai and Nanjing shows the low efficiency in $O_3$ production per molecule of $NO_x$ consumed. However, the OPE values in all six regions are generally lower than those in other remote and rural regions, in agreement with Wang et al. (2018), indicating that high precursor emissions in these regions are the main cause of high surface

$O_3$ concentrations.

## 6    Response of $O_3$ to emission controls

We quantify the response of daytime $O_3$ to emission changes to investigate the relationship between the chemical environment and the magnitude of $O_3$ changes for the six industrial regions of China. We implement three scenarios applying 20 % reductions in anthropogenic $NO_x$ emissions, VOC emissions and combined $NO_x$ and VOC emissions.

Spatial distributions of simulated daytime surface $O_3$ responses vary across China (Fig. 7). In the 20 % $NO_x$ emission control scenario, substantial $O_3$ increases (2-10 ppb) can be seen in the NCP, the YRD and the PRD, and $O_3$ concentrations decrease (0-8 ppb) in the Sichuan basin. In the 20 % VOC emission control scenario, there are small $O_3$ changes in most non-industrial regions of China (-1–2 ppb) but $O_3$ concentrations generally decrease by 1-9 ppb across the NCP, the YRD and the PRD. The Sichuan basin shows relatively small $O_3$ decreases. Areas showing $O_3$ increases in the 20 % $NO_x$ emission control experiment

match well with VOC limited areas indicated by the $NO_x$/VOCs and $H_2O_2$/$HNO_3$ ratios (cf. Fig. 5e, 5f vs Fig. 7a) suggesting that all the industrial regions considered here are VOC limited except Chongqing in the Sichuan basin that is $NO_x$ limited.

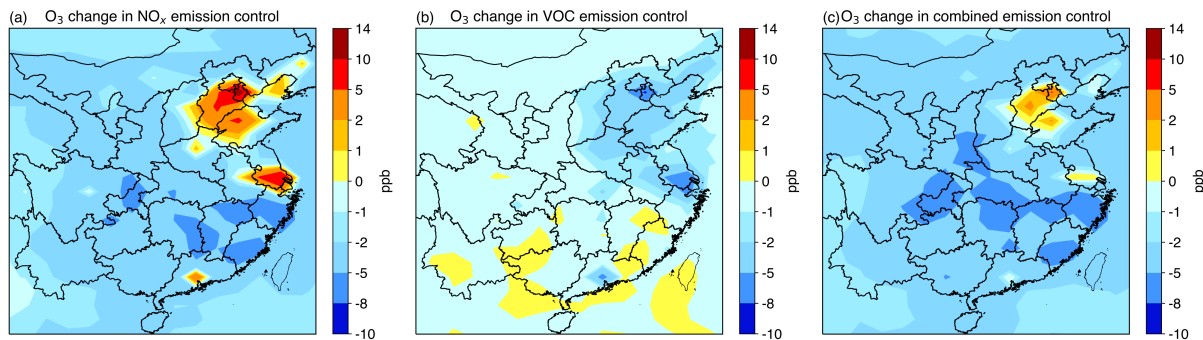

**Figure 7.** Spatial distributions of simulated surface daytime $O_3$ concentration changes (ppb) for **(a)** the 20 % $NO_x$ emission control, **(b)** the 20 % VOC emission control and **(c)** the 20 % combined $NO_x$ and VOC emission control compared to the present-day results in JJA, 2016, China.

In general, the greatest $O_3$ increases in the 20 % $NO_x$ control scenario occur in areas with high precursor concentrations. Shanghai shows the largest $O_3$ increases (11 %) (Table 4) and has the highest underlying $NO_x$ concentrations (Table 3). $O_3$ increases in Beijing and Guangzhou are similar (~8 %) possibly because of their similar $NO_x$ concentrations. Shijiazhuang

in the NCP shows the smallest $O_3$ increase (4 %) because of its lower $NO_x$ concentrations. In contrast, an $O_3$ decrease of



4 % is seen in Chongqing that is $NO_x$ limited. In the 20 % VOC control scenario, the largest $O_3$ decreases are simulated in Shanghai and Guangzhou (-10 %) while minimal $O_3$ decreases (-1 %) are simulated in Chongqing. The chemical environment in Chongqing is $NO_x$ limited and therefore the $O_3$ changes are not sensitive to VOC emissions.

**Table 4.** Simulated daytime mean $O_3$ concentrations and changes in $NO_x$, VOC and combined $NO_x$ and VOC emission controls for the six industrial regions in JJA, 2016, China.

| $O_3$ (ppb)/ Region | Base | $NO_x$ control | Change (%) | VOC control | Change (%) | $NO_x$+VOC control | Change (%) |
|---|---|---|---|---|---|---|---|
| Beijing | 78.0 | 84.7 | 8.6% | 72.5 | -7.0% | 79.7 | 2.2% |
| Shijiazhuang | 83.5 | 86.6 | 3.8% | 80.2 | -3.9% | 84.6 | 1.4% |
| Shanghai | 70.1 | 77.8 | 11.0% | 63.1 | -10.0% | 69.4 | -1.0% |
| Nanjing | 66.8 | 72.4 | 8.5% | 61.4 | -8.0% | 67.8 | 1.6% |
| Guangzhou | 60.2 | 64.8 | 7.6% | 53.8 | -10.7% | 60.4 | 0.3% |
| Chongqing | 93.8 | 89.5 | -4.6% | 92.5 | -1.4% | 88.5 | -5.6% |

In addition to separate 20 % reductions in $NO_x$ and VOC emissions, we demonstrate the importance of combined $NO_x$ and VOC emission controls to mitigate $O_3$ pollution in VOC limited regions. This effectively offsets the higher levels of $O_3$ that arise with $NO_x$ emission reductions alone. The $O_3$ increase in Shanghai is fully offset in the combined emission control (-1 %). While $O_3$ increases still occur in the other VOC limited regions, these increases are minimal (< 3 %). Reducing both $NO_x$ and VOC emissions decreases $O_3$ levels in Chongqing by 6 %. Therefore, combined emission controls are necessary to efficiently mitigate $O_3$ pollution in all these industrial regions, and VOC emission controls should be at least as stringent as $NO_x$ emission controls to address rising $O_3$ levels in these industrial regions.

## 7 Effectiveness of emission controls in reducing surface $O_3$ levels

To provide a more complete exploration of the effectiveness of emission controls, we construct a response surface of summertime daytime $O_3$ for each region to show the effect of changing $NO_x$ and VOC emissions. We do this by performing a set of 64 model simulations with global anthropogenic $NO_x$ and VOC emissions scaled independently over the range 0-140 % in increments of 20 %.

Figure 8 shows the magnitude and direction of $O_3$ changes in the six regions as $NO_x$ and VOC emissions change. For context, Fig. 8a also shows the simulated daytime $O_3$ changes between 2013 and 2019 in the Beijing region following estimated emission changes (Cheng et al., 2019). We find that simulated $O_3$ concentrations in Beijing increase from 71.6 ppb in 2013 to 82.6 ppb in 2019, an increase of 1.8 ppb/year. This is consistent with observed changes of 1.9 ppb/year over this period due to anthropogenic emission changes (Li et al., 2020). The observed daytime $O_3$ concentrations are 83 ppb in the Beijing region in 2019. This demonstrates that the model captures not only the magnitude and diurnal pattern of $O_3$ in summer 2016 well but also the observed $O_3$ changes in recent years.





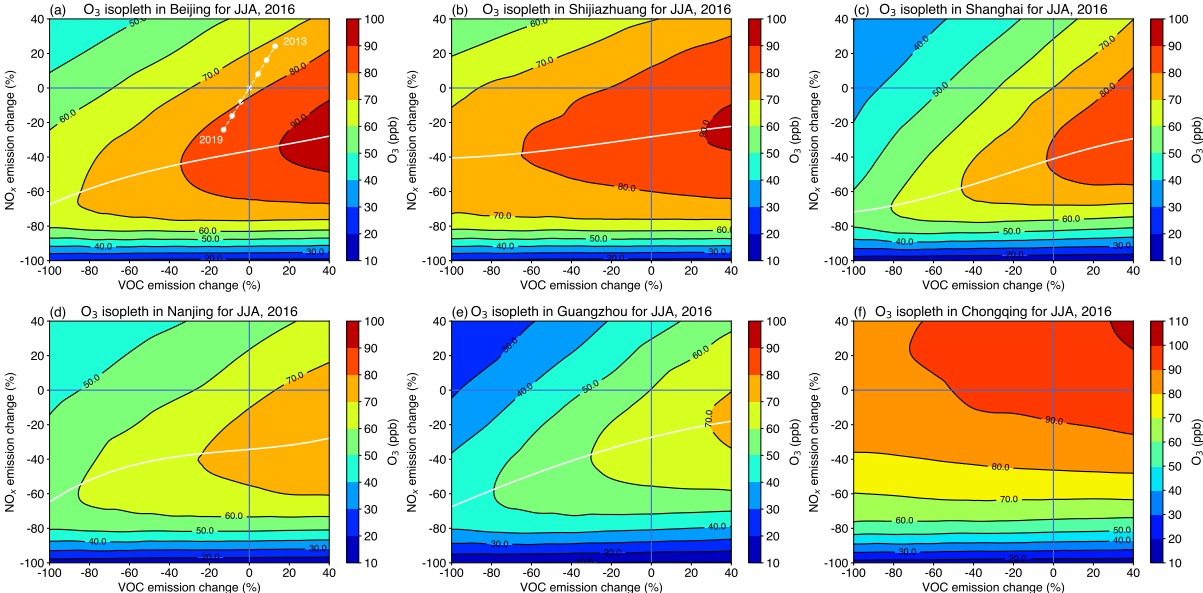

**Figure 8.** Simulated daytime surface $O_3$ responses (ppb) to anthropogenic $NO_x$ and VOC emission changes for the six industrial regions across China **(a – f)** in JJA, 2016. The intersection of the vertical and horizontal lines marks current $O_3$ levels. White ridge lines mark the peak in $O_3$ concentrations for given VOC emissions, and show the approximate transition between VOC limited (above the ridge) and $NO_x$ limited (below the ridge) regimes. White dots in **(a)** represent simulated daytime $O_3$ levels in the Beijing region in JJA between 2013 and 2019 following estimated $NO_x$ and VOC emission changes.

The patterns of $O_3$ response seen in the VOC limited regions (Fig. 8a-e) are similar, such that decreases in $NO_x$ emissions from their current levels increase $O_3$ concentrations. Large $O_3$ increases occur in Shanghai and Beijing, highlighting that it is

not beneficial to reduce $NO_x$ emissions unless VOC emissions are also reduced. Large reductions (~40 %) in $NO_x$ emissions are required to shift the chemical environment from VOC limited to $NO_x$ limited for these two regions. The large decrease in $O_3$ in Shanghai and Guangzhou when reducing VOC emissions indicates that the efficiency in lowering $O_3$ levels by decreasing VOC emissions is high in these regions. In contrast, the efficiency of VOC emissions alone in reducing $O_3$ levels is lower in Shijiazhuang and Chongqing.

Figure 9 shows the $O_3$ responses in each region to changes in $NO_x$ emissions, VOC emissions and combined $NO_x$ and VOC emissions, which represent cross-sections through the $O_3$ response surfaces shown in Fig 8. It is difficult to decrease $O_3$ concentrations in Shanghai by reducing $NO_x$ emissions alone because there is a steep rise in surface $O_3$ (~15 %) when $NO_x$ emissions are reduced by 40 % from the current state. Decreasing $O_3$ from current levels requires reductions in $NO_x$ emissions of more than 50 % for Shijiazhuang and Guangzhou and more than 70 % for Beijing, Shanghai and Nanjing. This suggests that

mitigating poor $O_3$ air quality in these VOC limited regions through $NO_x$ emission controls alone would require much greater reductions than the 21 % reductions in $NO_x$ emissions that are reported to have occurred in China from 2013 to 2017 (Zheng et al., 2018).





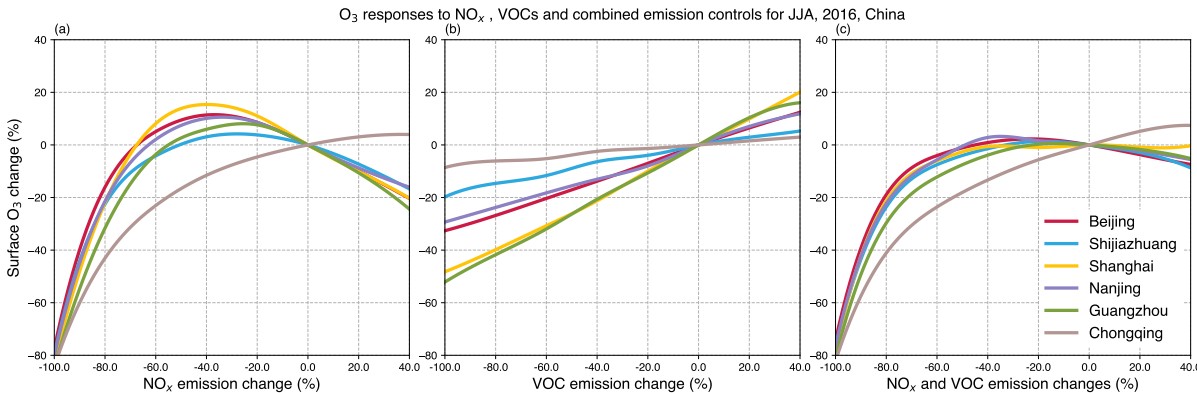

**Figure 9.** Simulated daytime surface $O_3$ responses to changes in anthropogenic emissions of **(a)** $NO_x$, **(b)** VOC and **(c)** combined $NO_x$ and VOC emissions for the six industrial regions in JJA, 2016, China.

$O_3$ responses to VOC emission changes are smaller and more linear than the responses seen for $NO_x$ emissions changes (Fig. 9a, 9b). Reducing VOC emissions by 40 % gives large decreases in $O_3$ concentrations (20 %) in Shanghai and Guangzhou and

smaller decreases (< 10 %) in Shijiazhuang and Chongqing (Fig. 9b). Reductions in VOC emissions are key to reducing present-day $O_3$ concentrations as they effectively offset the rising $O_3$ levels due to decreasing $NO_x$ emissions (Fig. 9c). Emission reductions of 50 % or more are required to reduce $O_3$ levels for all regions if controls on $NO_x$ and VOC emissions are applied simultaneously.

To place our results in a wider global context, Figure 10 shows summer-mean surface $O_3$ changes over different regions

with high emissions in other parts of the world compared with those in China. We consider six major industrialised regions outside of China and select the model grid-cell that is most closely co-located with the region. We note that proportional increases in summer-mean $O_3$ are larger than that of daytime $O_3$ increases when $NO_x$ emissions are reduced (see Fig. 9), principally because absolute $O_3$ concentrations are smaller with the inclusion of nighttime conditions. We find that all selected high emission regions across the globe outside of China are $NO_x$ limited, such that $NO_x$ emissions decreases yield regional

$O_3$ decreases. Current levels of $NO_x$ emissions in these regions are considerably lower than for the industrial regions of China, reflecting the different $O_3$ sensitivity regimes (Table 5).



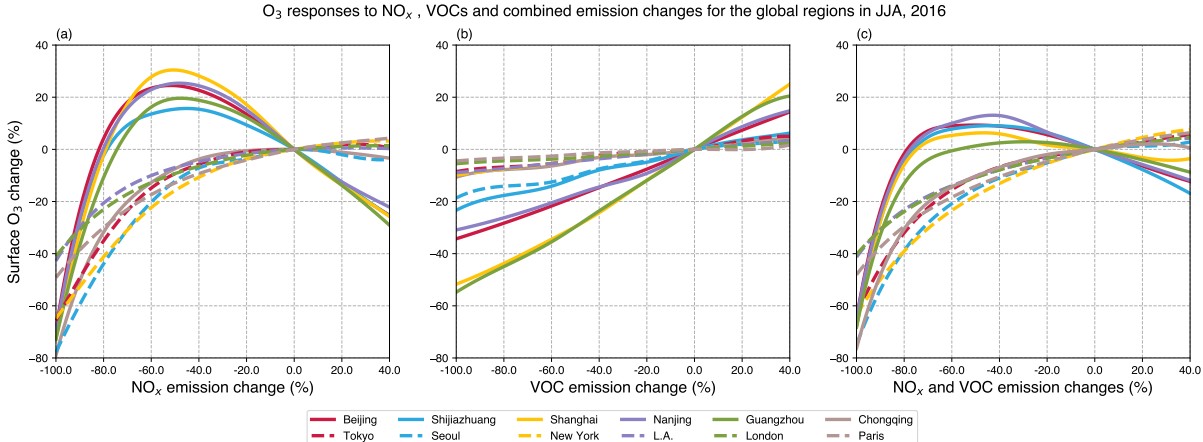

**Figure 10.** Simulated summertime mean surface $O_3$ responses to anthropogenic **(a)** $NO_x$, **(b)** VOC and **(c)** combined $NO_x$ and VOC emission changes in regions across the globe: Tokyo, Seoul, New York, L.A., London, Paris (dashed lines) and those in major industrial regions of China (solid lines) in JJA, 2016.

Reductions of both $NO_x$ and VOC emissions substantially decrease $O_3$ levels for these selected regions outside of China, and the magnitude of the $O_3$ decreases are similar to those found for Chongqing (Fig 10). Conversely, the magnitude of $O_3$ decreases when reducing VOC emissions are smaller than all five VOC limited regions in China. This indicates that $O_3$
concentrations are less sensitive to VOC emissions in these other world regions due to their lower VOC emissions (Table 5).
Despite lower $NO_x$ and VOC emissions in the regions outside of China, surface $O_3$ concentrations, particularly in the Seoul and New York regions, are similar to those for China. This highlights that regional $O_3$ levels also depend on background $O_3$ concentrations, despite localised $NO_x$ and VOC emissions that lead to different $O_3$ production regimes. The $O_3$ levels in European regions e.g. London and Paris are lowest, in accordance with the lowest $NO_x$ and VOC emission levels. Overall,
these results show that there are substantial differences in the efficiency of emission control scenarios to reduce surface $O_3$ levels in different parts of the world. For many industrial regions of China, the extended regions are VOC limited and hence, reductions of VOC emissions are the key to reducing poor $O_3$ air quality. For other regions selected in this study $NO_x$ emission reductions are still pertinent to improving $O_3$ pollution.





**Table 5.** Anthropogenic $NO_x$ and VOC emissions ($*e^{-10}$ kg m$^{-2}$ s$^{-1}$) and summertime mean surface $O_3$ concentrations (ppb) in regions across the industrial regions of China and the globe. MEIC emissions of 2013 adjusted for 2016 are used for Chinese industrial regions. HTAP emissions of 2010 are used for other regions of the globe.

| Region | $NO_x$ emissions | VOC emissions | $O_3$ conc. |
|---|---|---|---|
| **China** | | | |
| Beijing | 5.5 | 6.7 | 43.4 |
| Shijiazhuang | 4.2 | 4.6 | 47.6 |
| Shanghai | 7.4 | 9.6 | 34.4 |
| Nanjing | 6.9 | 8.1 | 35.9 |
| Guangzhou | 8.4 | 12.0 | 28.0 |
| Chongqing | 3.1 | 3.6 | 56.0 |
| **Global** | | | |
| Tokyo | 2.0 | 2.6 | 38.9 |
| Seoul | 1.5 | 2.1 | 45.5 |
| New York | 2.3 | 3.1 | 45.3 |
| L.A. | 1.1 | 1.3 | 40.1 |
| London | 1.1 | 1.5 | 30.6 |
| Paris | 0.8 | 1.0 | 32.6 |

## 8 Conclusions

This study presents the application of the global chemistry-climate UKCA model with an improved gas-phase chemistry scheme including more reactive VOCs to simulate regional summertime $O_3$ pollution across major industrialised regions in China for the first time. Differences in atmospheric chemical environments are investigated, and the effectiveness of different emission control strategies in reducing $O_3$ concentrations is quantified. The model captures the magnitude, diurnal profiles and diurnal variation of $O_3$ concentrations across most industrial regions well. We highlight that peak $O_3$ concentrations can be 325 captured well, indicating that $O_3$ production can be effectively simulated with more highly active VOC oxidation environments for high emission regions of China.

Simulated daytime $O_3$ levels are highest on the North China Plain (Beijing and Shijiazhuang), and in the Sichuan Basin (Chongqing), and are lowest in the Pearl River Delta (Guangzhou). We find that there is a systematic bias in $O_3$ throughout the diurnal cycle in Chongqing reflecting the mountainous inland area that is inadequately captured by the topography in the 330 model. The $O_3$ production rates are highest in the Pearl River Delta compared to other regions. However, its much lower $O_3$ levels reflect the importance of meteorological impacts in this coastal region. OPE values in these industrial regions are low, indicating that their high $O_3$ levels are mainly caused by high precursor emissions. Both $O_3$ sensitivity ratios we apply here ($NO_x$/VOCs and $H_2O_2$/$HNO_3$) suggest that all the industrial regions except Chongqing are VOC limited.





A set of simulations are performed with a range of $NO_x$ and VOC emissions to construct $O_3$ response surfaces to assess
the impacts of different emission control strategies in different regions. Reducing $NO_x$ emissions alone by 20 % leads to a
substantial $O_3$ increase (11 %) in Shanghai. Reductions in VOC emissions alone of 20 % produce the largest decrease (-
11 %) in $O_3$ levels in Shanghai and Guangzhou and the smallest decrease (- 1 %) in Chongqing. We find that reducing $O_3$
concentrations across all industrial regions of China would require more than 70 % reductions if reducing $NO_x$ emissions
alone, and therefore VOC emission controls are important to reduce $O_3$ levels. We also find that combined emission controls
effectively offset high $O_3$ levels that arise from reduced $NO_x$ emissions alone. These responses are substantially different
from those currently found in major highly populated regions in other parts of the world. The results show $NO_x$ limited $O_3$
production in these global areas, which also reflects the predominance of heavily populated VOC limited areas across the
industrial regions in China. Therefore, $O_3$ pollution in the industrial regions of China should be treated as a regional issue and
regional VOC emission control strategies should be considered.

The new capabilities for simulating regional surface $O_3$ pollution developed here will be helpful for future model studies
to investigate the regional $O_3$ impacts on climate. The magnitude of $O_3$ changes over recent years in the Beijing region can
be reproduced well. There remain model biases in regions with complex topography and high elevation – a common issue for
global and regional models. Another source of uncertainty is the rapid change in anthropogenic emissions in recent years in
China, which presents a particular challenge for inventory development. Recently, while $NO_x$ emissions have been successfully
reduced across many regions in China, changes in VOC emissions have been relatively small, and this has led to an increase in
$O_3$ concentrations in many regions. Regional VOC emission controls are hence urgently needed to maximise the effectiveness
in reducing surface $O_3$ pollution in China.

*Data availability.*    The data generated in this study will be made available through the Edinburgh DataShare at https://datashare.is.ed.ac.uk/handle/xxxxx.

*Author contributions.*    ZL, RD and OW designed the study. ZL, MH and FO'C set up the model. ZL ran model simulations and performed
the analysis. ZL, RD and OW prepared the paper with contributions from all co-authors.

*Competing interests.*    The authors declare that they have no conflict of interest.

*Acknowledgements.*    ZL thanks the University of Edinburgh China Scholarship Council. MH, OW and RD thank the Natural Environment
Research Council (NERC) for funding under grants NE/N006925/1, NE/N006976/1 and NE/N006941/1. This work made use of computation
resources on the Met Office and NERC joint supercomputer system (MONSooN) in the UK. ZL thanks the UKCA community for help in
the model set-up.



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
