# Peer review of "Contrasting chemical environments in summertime for atmospheric ozone across major Chinese industrial regions: the effectiveness of emission control strategies"

_Atmospheric Chemistry and Physics, 2020_

## Referee Comment (RC2)

This manuscript investigated chemical environment for surface O3 for six major industrial regions across China in summer 2016. Detailed chemistry-climate model simulations were employed to diagnose ozone sensitivity to precursors and contrast the effectiveness of different measures to reduce surface O3 concentrations. This manuscript is helpful to understand ozone pollution mechanism in Chinese cities, and within the scope of ACP. I think it is publishable in ACP after my following concerns are addressed.

Line 215: The gross rate of production $P(O_3)$ actually represents the production rate of $O_x$ ($O_3$ + $NO_2$) through the reaction $HO_2$ ($RO_2$) +NO. Therefore, the net ozone production rate should include the loss term $NO_2$+OH (Wang et al., 2019. doi.org/10.5194/acp-19-9413-2019). In addition to $OH+NO_2$ and $RO_2+NO_2$, the loss of $NO_x$ should also include $RO_2$+NO and OH+HONO When calculating OPE. Please give specific quantification even though these reactions play a minor role in the loss of $NO_x$,.

Figure 4 shows significant underestimation for $NO_2$ in daytime, but overestimation for NO2 at nighttime. The overestimation of $NO_2$ at night maybe related to underestimated nighttime chemistry such as the removal of NO3 and N2O5 through heterogenous uptake (Li et al., 2018;Li et al., 2019). A short discuss should be performed. Additionally, how do these underestimation and overestimation for $NO_2$ influence your diagnosis of ozone sensitivity? For example, the underestimation of $NO_2$ in Chongqing will lead to more $NO_x$-limited, which likely misleads the actual situation.

Figure 8. shows ozone increased from 70 ppb to over 80 ppb during 2013-2019. However, observed ozone concentrations in Beijing didn't increased significantly during the period or decreased after 2015 in spite that ozone increased over North China Plain (Lu et al., 2018. DOI: 10.1021/acs.estlett.8b00366; Tang et al., 2020. doi.org/10.1016/j.atmosres.2020.105333). This needs further explanations.

Line 270: How do you obtain VOC and $NO_x$ emissions in 2018 and 2019 given that Cheng et al (2019) just estimated emissions during 2013-2017. Please give specific description.

Line 145: There are only 450 measurement stations in 2013, growing to 1,500 stations in 2017 and 1670 stations in 2019.
Line 300: "summer-mean ozone" should be "daily mean ozone".

references
Li, J., Chen, X., Wang, Z., Du, H., Yang, W., Sun, Y., Hu, B., Li, J., Wang, W., and Wang, T.: Radiative and heterogeneous chemical effects of aerosols on ozone and inorganic aerosols over East Asia, Science of the Total Environment, 622, 1327-1342, 2018.
Li, K., Jacob, D. J., Liao, H., Zhu, J., Shah, V., Shen, L., Bates, K. H., Zhang, Q., and Zhai, S.: A two-pollutant strategy for improving ozone and particulate air quality in China, Nature Geoscience, 12,

906–910, 10.1038/s41561-019-0464-x, 2019.

---

## Author Comment (AC1)

Dear editor and all reviewers:

We thank the editor and all reviewers for their contribution to the improvement of the ACP manuscript. Responses to reviewers on "Contrasting chemical environments in summertime for atmospheric ozone across major Chinese industrial regions: the effectiveness of emission control strategies" by Zhenze Liu et al. are given below. For clarity, the reviewer comments are given in bold, followed by our responses and modified text in our revised manuscript are given in quotes, italics and blue.

**Response to Reviewer 1:**

1. In this study the UKCA chemistry-climate model is enhanced by incorporating reactive VOC tracers into the UKCA gas-phase chemistry scheme in order to better represent urban and regional-scale O3 photochemistry, and then applied to quantify the differences in chemical environment for surface O3 for six major industrial regions across China in summer 2016. This study is well organized and clearly written on an interesting topic – how to effectively control tropospheric ozone pollution in China. Here are some specific comments.

We thank the reviewer for their positive comments here, and address specific concerns below.

2. Six cities are chosen because they are located in the heavily populated regions with high emissions, but their climate is different, for example, Chongqing is often cloudy and foggy, and especially hot and muggy in summer. The amount of sunshine in Chongqing is poor, and the only months when the sunshine hours exceed 40% (but they still remain below 50%) are July and August; while Beijing has strange weather—extreme hot and cold temperatures, and high humidity to no humidity. It has glorious sunshine year-round and is hot and rainy in summer, especially in July and August. In the study impacts of meteorological conditions on local O3 production rates are not discussed, and model results are not evaluated against observed meteorological parameters.

We thank the reviewer for this comment and have investigated meteorological conditions further. We agree that meteorological conditions are important for  $O_3$  formation, and many studies have focused on this aspect specifically (Gong and Liao, 2019; Liu and Wang, 2020; Shi et al., 2020). We have added additional text to introduce the importance of meteorology to  $O_3$  formation in the introductory section of the paper:

Page 2, line 47:

"Meteorological processes also affect  $O_3$  formation through temperature, humidity, clouds, precipitation and biogenic emissions, and a number of papers have studied meteorological impacts on  $O_3$  over China (Gong and Liao, 2019; Liu and Wang, 2020; Shi et al., 2020). However, emission controls are the primary strategies used to reduce

**$O_3$ pollution and we focus on these for this study, as their effectiveness for different regions has not been fully investigated."**

We emphasise that the model is nudged to ECMWF ERA-interim meteorological reanalysis data, hence the meteorology over these broad regions is generally simulated well. We show a meteorological evaluation for temperature and humidity for the regions we consider in Table 1. Typically, high temperature and low humidity benefit O3 production in China, as outlined in the studies listed above. Observed daily mean data have been obtained from the national meteorological data center http://data.cma.cn/.

|              | Temperature (°C) |      |      |      |     |  |
|--------------|------------------|------|------|------|-----|--|
|              | Obs.             | Sim. | Bias | RMSE | r   |  |
| Beijing      | 24.0             | 21.7 | -2.3 | 2.7  | 0.8 |  |
| Shijiazhuang | 26.3             | 24.3 | -2.0 | 2.8  | 0.7 |  |
| Shanghai     | 27.6             | 25.6 | -2.0 | 2.6  | 0.9 |  |
| Nanjing      | 27.8             | 26.9 | -0.9 | 2.0  | 0.9 |  |
| Guangzhou    | 28.3             | 27.3 | -1.0 | 1.8  | 0.5 |  |
| Chongqing    | 29.4             | 27.5 | -1.9 | 3.0  | 0.8 |  |

**Table 1:** Comparison of daily mean of observed and simulated surface temperature (°C) and relative humidity (%) for six regions of China during June, July, August, 2016.

|              | Relative Humidity (%) |      |      |      |     |  |  |
|--------------|-----------------------|------|------|------|-----|--|--|
|              | Obs.                  | Sim. | Bias | RMSE | r   |  |  |
| Beijing      | 67.0                  | 79.6 | 12.6 | 14.6 | 0.8 |  |  |
| Shijiazhuang | 72.1                  | 80.8 | 8.7  | 13.2 | 0.8 |  |  |
| Shanghai     | 80.4                  | 79.8 | -0.6 | 5.8  | 0.8 |  |  |
| Nanjing      | 77.3                  | 77.9 | 0.6  | 5.5  | 0.9 |  |  |
| Guangzhou    | 85.5                  | 82.8 | -2.7 | 6.0  | 0.6 |  |  |
| Chongqing    | 68.2                  | 81.4 | 13.2 | 16.5 | 0.5 |  |  |

From Table 1, we can see that the model is biased low for simulated temperature, although this is partly due to use of temperature in the lowest model layer rather than at 2 m altitude. However, the model successfully captures regional differences in temperature – the highest in Chongqing and the lowest in BTH regions e.g. Beijing and Shijiazhuang. Relative humidity in Shanghai, Nanjing, Guangzhou is relatively well simulated; the highest relative humidity for Guangzhou is also captured by the model. The model biases for Beijing and Chongqing are larger, but this is likely to be due to their proximity to mountainous areas influencing the respective model grid boxes. Higher humidity promotes O3 destruction hence may lead to lower surface O3 levels, and this could partly explain negative model biases in daily mean surface O3 levels for Beijing. However, we note that temperature and humidity are not the main influences on surface O3 concentrations in Chongqing, and topography is a more important factor as discussed in the paper. Given our use of reliable ECMWF ERA-interim meteorological reanalysis data and the uncertainties associated with meteorological

evaluation, we choose not to include this discussion in the revised manuscript. For clarity we have added the following text to section 3:

Page 11, line 231:

"Given our use of reliable meteorological reanalysis data, we note that meteorology is not the main influence on the model biases. We therefore investigate O3 chemical environments in different regions to explore regional differences below."

3. Figure 3 shows that the model overestimates O3 systematically in Chongqing, and observed low concentrations (<35 ppb) are not reproduced. As statistical indicators are limited and independent in this study, high correlation coefficient r does not indicate that the modeled concentrations in Chongqing are acceptable, because seasonal averaged diurnal variations in O3 over high emission areas are quite easy to be simulated. Beside the explanation in lines 165-169, would the meteorological fields be attributable?

Chongqing is not simulated as well as other regions as we note in the text. The overestimation of surface  $O_3$  levels for Chongqing is likely linked to the complex topography as discussed in section 3. Surface  $O_3$  in Chongqing is representative of higher surface altitudes characteristic of the region, leading to a systematic high bias compared with observations, and a corresponding low bias for NO2 concentrations. Our results suggest that the region of Chongqing represents a NOx limited regime, hence the underestimated NO2 levels is not the cause to the overpredicted O3 levels. This is consistent with satellite-based assessments of ozone sensitivity in China for 2016 (Wang et al., 2021). It hence suggests that complex transport patterns associated with the topography and the higher altitude of the region are the most important factors influencing O3 levels rather than meteorology as discussed above. We have added the following text to section 3:

Page 11, line 236:

"This underestimation may lead to overestimated  $O_3$  concentrations in a VOC-limited regime and underestimated  $O_3$  in a NOx-limited regime. While underestimated NOx concentrations may reflect underestimated NOx emissions, it is more likely to arise from the effects of dilution on NOx."

**4. From Figure 4, we can also find that NO2 in Chongqing is not reproduced well, and daytime NO2 is underestimated at all sites. How would the underestimation of daytime NO2 impact on O3 production sensitivity?**

We agree that the underestimation of daytime  $NO_2$  concentrations will influence  $O_3$  sensitivity. However, higher  $NO_x$  levels lead to a more VOC limited regime so all regions except Chongqing would still be in VOC limited regimes if  $NO_x$  levels were increased. From the  $O_3$  isopleth plot in Fig 8, we can see that increasing  $NO_x$  levels would leave Chongqing in the transition regime or still in a  $NO_x$  limited regime.

However, to address this concern we have toned down our statement that 'Chongqing is  $NO_x$  limited' to accommodate the local underestimation of  $NO_2$  concentrations. Text added in section 6 and 7 in the revised manuscript for clarity are as follows:

**Page 17, line 348:**

"We note that our conclusion of  $NO_x$  limitation in Chongqing may be sensitive to our underestimation of  $NO_2$  levels (section 3), and that with higher  $NO_2$  Chongqing may shift from the  $NO_x$  limited regime to the VOC limited regime. However, satellite observation based studies have also identified this region as one that is largely  $NO_x$ limited, in contrast to the heavily populated coastal regions (Wang et al., 2021)."

**Page 19, line 397:**

"From the O3 isopleth for Chongqing, we can see that even if  $NO_x$  emissions were increased by 40 %, Chongqing would still be in  $NO_x$  limited or transition regimes. This suggests that Chongqing is still far away from a VOC limited regime."

**5. Bye the way, O3 and NO2 from the surface monitoring networks of China are usually recorded in the unit of µg/m3, how are they converted to ppb?**

The units of  $\mu g/m^3$  are converted to ppb based on temperature and pressure as follows:

 $\mu g/m^3 = ppb * molecular weight/molecular volume (litres)$

where molecular volume = 22.41 \* (temperature/273.15) \* (1013/pressure)

- T = absolute temperature (K)
- P = atmospheric pressure (hPa)

Chemical species in units of mixing ratio (kg/kg) are output by the model and then converted to ppb. This unit conversion is a standard procedure and so we do not include a description of the approach in the paper.

6. Tropospheric O3 concentrations are functions of the chain lengths of NOx and Hox radical catalytic cycles, and ozone production rates depend on not only NOx and VOCs concentrations, but also actinic flux and temperature. The conclusion of O3 production across all regions except Chongqing being VOC limited needs to studies for different weather conditions.

 $NO_x$  and VOCs are the main primary O3 precursor species emitted from emission sources. Therefore, we construct O3 response surfaces with respect to changing NOx and VOC emissions, and our aim is to investigate the impacts of changing NOx and VOC emissions on surface O3 concentrations. We choose to represent O3 sensitivity to NOx and VOC emissions because it is largely driven by ambient abundances of NOx and VOCs – reflected by VOC and NOx limited regimes. We agree that actinic flux and temperature alter O3 production rates through their effects on photolysis rates and chemical reaction rates, and these are fully captured in our model chemistry scheme, which includes 101 species and 348 reactions (see section 2.1). We note that O3 sensitivity is affected by photolysis rates and temperature, but highlight that we use the meteorological conditions appropriate to each region to calculate the sensitivity. It is a good point that O3 sensitivity could be investigated in future studies for different weather conditions, and it is also worthy investigating how O3 sensitivity changes throughout a day. However, this study aims to provide a broad assessment of regional O3 sensitivity with the implication of emission control strategies in the summer time, and focus on specific time periods or unusual meteorological conditions is not necessary for this.

**7. Generally model simulated O3 production rates are quite sensitive to the chemical mechanism beside other model inputs. As stated in lines 101-105, VOCs such as alkenes and aromatics are abundant in industrial areas of China, but why is ethene not incorporated into the UKCA gas-phase chemistry scheme?**

The extended chemistry scheme for UKCA introduced in this study provides highly reactive VOC species to permit representation of more active photochemical environments. We incorporate  $C_3H_6$ ,  $C_4H_{10}$  and toluene as proxies for the principal chemically reactive VOC families: alkenes, alkanes and aromatics. The tracer  $C_3H_6$  thus implicitly includes ethene as a part of the alkene family. This approach to VOC chemistry is described in detail in Stockwell et al. (1990). We highlight that this extended chemistry scheme permits a more realistic VOC oxidation environment, and is more suitable for high emission areas than the original chemistry scheme that did not include these short-lived VOC tracers.

**Response to Reviewer 2:**

1. This manuscript investigated chemical environment for surface O3 for six major industrial regions across China in summer 2016. Detailed chemistryclimate model simulations were employed to diagnose ozone sensitivity to precursors and contrast the effectiveness of different measures to reduce surface O3 concentrations. This manuscript is helpful to understand ozone pollution mechanism in Chinese cities, and within the scope of ACP. I think it is publishable in ACP after my following concerns are addressed.

We thank the reviewer for their positive comments, and specific concerns are addressed below.

2. Line 215: The gross rate of production  $P(O_3)$  actually represents the production rate of  $O_X$  ( $O_3 + NO_2$ ) through the reaction  $HO_2$  ( $RO_2$ ) +NO. Therefore, the net ozone production rate should include the loss term  $NO_2$ +OH (Wang et al., 2019. doi.org/10.5194/acp-19-9413-2019). In addition to OH+NO2 and  $RO_2$ +NO2, the loss of NOx should also include  $RO_2$ +NO and OH+HONO When calculating OPE. Please give specific quantification even though these reactions play a minor role in the loss of NOx.

We thank the reviewer for this point. The calculation of O3 production that the reviewer points out is based on including NO2 in the definition of odd oxygen (Ox). We note that there are different definitions of Ox and sometimes O3 and NO2 are together defined as Ox. These wider definitions include O3, O(1D), O(3P), NO2 and various other NOy species (NO3, N2O5, HO2NO2, HNO3, PAN, MPAN), see Section 4 in Horowitz et al. (2003). There is still ongoing debate about the best way to consider Ox, and recent studies have used an even wider definition (Bates and Jacob, 2020). However, we choose to take a simpler and more traditional definition of Ox including only O3 and O species, following Kleinman et al. (1997). No approach provides a complete assessment of all odd oxygen, but we feel strongly that the simplicity and greater interpretability of this more traditional approach outweighs the benefits of including a large number of minor species, while also allowing direct comparability with earlier studies which have used the same approach. Using this approach, O3 production is the sum of HO2/RO2 + NO reactions and O3 loss is the sum of direct reactions of O3 with OH, HO2 and alkenes, and O(1D) + H2O. We now state this clearly in section 5:

Page 15, line 298:

"We define the net O3 production rate (ppb/h) as the gross rate of production of O3, P(O3), from the reactions HO2 + NO and RO2 + NO minus the gross rate of loss of O3,  $L(O_3)$ , from the reactions  $O(^1D) + H_2O$ ,  $O_3 + OH$ ,  $O_3 + HO_2$  and  $O_3 + VOCs$ ."

We regard to the reaction  $NO_2 + OH$  as the principal loss of  $NO_x$  but it is not a loss of  $O_3$  because we do not include  $NO_2$  as part of odd oxygen – this is an advantage of the

simpler definition of  $O_x$ . The NOx loss term is defined as NO2 + OH in Liu et al. (1987), assuming that the lifetime of NOx is determined by the formation and deposition of HNO3 in the daytime. However, the formation of organic nitrate (RONO2) is also a minor loss of NO2 as the reviewer points out. We therefore include this reaction in the NOx loss term as the reviewer suggests. It also has minor effects on the calculation of OPE. To account for the reviewer's suggestion, we have redefined the NOx loss and OPE in the paper and have updated Fig. 6. Our results in the text are unchanged. We have also introduced RONO2 in the introductory section. Oxidation of HONO by OH does not remove NOx so this reaction is not included in the NOx loss term. The modified text is shown below.

**Page 2, Line 60:**

"However, at high NOx concentrations, nitric acid (HNO3), peroxy nitrates (RO2NO2) and organic nitrates (RONO2) are easily formed as NOx reacts with OH and RO2. These species are the main sinks of radicals and NOx, and are readily removed from the atmosphere by deposition or exported to remote areas (Horowitz et al., 1998). Therefore, increasing NOx concentrations increase O3 production, but also accelerate the formation of NOx sinks, leading to less efficient O3 formation."

**Page 15, Line 308:**

"The loss of  $NO_x$ ,  $L(NO_x)$ , is principally determined by the reactions  $OH + NO_2$ ,  $RO_2 + NO_2$  and  $RO + NO_2$ , which produce  $HNO_3$ ,  $RO_2NO_2$  and  $RONO_2$  respectively."

**Figure 6:** Simulated surface daytime (a) net  $O_3$  production rates, gross  $O_3$  production rates and gross  $O_3$  loss rates (ppb/h) (b) gross  $O_3$  production rates and NOx loss rates (ppb/h) (c) OPE (unitless) for the six industrial regions in JJA, 2016, China.

3. Figure 4 shows significant underestimation for NO2 in daytime, but overestimation for NO2 at nighttime. The overestimation of NO2 at night maybe related to underestimated nighttime chemistry such as the removal of NO3 and N2O5 through heterogenous uptake (Li et al., 2018; Li et al., 2019). A short discuss should be performed. Additionally, how do these underestimation and overestimation for NO2 influence your diagnosis of ozone sensitivity? For example, the underestimation of NO2 in Chongqing will lead to more NOx-

**limited, which likely misleads the actual situation.**

The heterogeneous chemistry scheme in UKCA includes heterogenous uptake for NO3 and  $N_2O_5$  – the conversion of  $N_2O_5$  to aqueous HNO3. Uptake coefficients are estimated for the different aerosol types included in the GLOMAP aerosol scheme. In UKCA, the lack of nitrate aerosol in the aerosol scheme may result in the lower uptake of nitrogen (Archibald et al., 2020), particularly in regions with high NOx emissions. Therefore, the heterogeneous removal of nitrogen may be biased, potentially leading to higher NO2 and lower O3 concentrations at nighttime. We have now added further text to discuss the potential impacts of heterogeneous processes in section 3 on model evaluation:

**Page 7, Line 202:**

"In addition, nighttime heterogeneous uptake of nitrogen on aerosols remains highly uncertain due to the complexity in estimating uptake coefficients for different aerosol composition/mixing states (Lowe et al., 2015; Tham et al., 2018). In UKCA, the lack of nitrate aerosol in the aerosol scheme may result in a lower uptake of nitrogen (Archibald et al., 2020), particularly in regions with high  $NO_x$  emissions. Therefore, there may be a bias in the heterogeneous removal of nitrogen, potentially leading to higher  $NO_2$  and lower  $O_3$  concentrations at nighttime."

However, we note that we focus on daytime O3 levels and O3 sensitivity, and therefore uncertainties in nighttime chemistry have a relatively small influence on our results.

As in our response to the reviewer 1, we now include discussion of the impacts of underestimated NO2 levels on O3 sensitivity. We note that higher NO2 levels for Beijing, Shijiazhuang, Shanghai, Nanjing, Guangzhou would lead to more VOC limited chemical environments, so this would not affect our conclusions. However, it is possible that higher NOx emissions might shift Chongqing into a transition regime (see Fig 8 in section 6). We have adjusted the statement that 'Chongqing is NOx limited' in sections 6 and 7 as follows:

**Page 17, line 348:**

"We note that our conclusion of  $NO_x$  limitation in Chongqing may be sensitive to our underestimation of  $NO_2$  levels (section 3), and that with higher  $NO_2$  Chongqing may shift from the  $NO_x$  limited regime to the VOC limited regime. However, satellite observation based studies have also identified this region as one that is largely  $NO_x$ limited, in contrast to the heavily populated coastal regions (Wang et al., 2021)."

**Page 19, line 397:**

"From the O3 isopleth for Chongqing, we can see that even if  $NO_x$  emissions were increased by 40 %, Chongqing would still be in  $NO_x$  limited or transition regimes. This suggests that Chongqing is still far away from a VOC limited regime." 4. Figure 8. shows ozone increased from 70 ppb to over 80 ppb during 2013-2019. However, observed ozone concentrations in Beijing didn't increased significantly during the period or decreased after 2015 in spite that ozone increased over North China Plain (Lu et al., 2018. **DOI**: 10.1021/acs.estlett.8b00366; Tang 2020. et al., doi.org/10.1016/j.atmosres.2020.105333). This needs further explanations.

Tang et al. (2021) suggest that the chemical environment for Beijing is now NOx limited hence reductions of NOx emissions lower surface O3 levels. However, our study focuses on the regional scale situation rather than on urban measurement locations. Many studies have shown an overall increasing trend in O3 levels on the North China Plain in recent years, as the reviewer notes. It would not be representative to use a single measurement site in the outskirts of the city to interpret the overall trend of the whole region. We have added text in the conclusion section to emphasise the aim of this study:

Page 23, line 476:

"This study hence provides a broad assessment of the  $O_3$  sensitivities for these regions with the implication of emission control strategies."

**5. Line 270: How do you obtain VOC and NOx emissions in 2018 and 2019 given that Cheng et al (2019) just estimated emissions during 2013-2017. Please give specific description.**

We assume that  $NO_x$  and VOC emissions in the Beijing region continue to follow the same trend as they do between 2013 and 2016. We have now added further text to clarify this in section 7. We already note in the conclusions that emission inventory development is a limitation for accurately predicting O3 trends.

Page 18, line 383:

"For context, Fig. 8a also shows the simulated daytime  $O_3$  changes between 2013 and 2019 in the Beijing region assuming that the emission changes observed between 2013 and 2016 continue at the same rate until 2019 (Cheng et al., 2019)."

**6. Line 145: There are only 450 measurement stations in 2013, growing to 1,500 stations in 2017 and 1670 stations in 2019.**

Yes, the number of measurement sites in China increased substantially after 2013. The text in the paper hase been modified to reflect this.

Page 6, Line 177:

"450 measurement stations in China started operating in 2013, growing rapidly to 1670 stations by 2019."

**7. Line 300: "summer-mean ozone" should be "daily mean ozone".**

We replace 'summer-mean ozone' with 'summer daily-mean ozone' throughout the text in section 7.

Below are all references used in the responses. Added references in the revised manuscript are shown as track changes.

**References:**

[revised manuscript text omitted]

---

## Author Response (AR1)

Dear editor and all reviewers:

We thank the editor and all reviewers for their contribution to the improvement of the ACP manuscript. Responses to reviewers on "Contrasting chemical environments in summertime for atmospheric ozone across major Chinese industrial regions: the effectiveness of emission control strategies" by Zhenze Liu et al. are given below. For clarity, the reviewer comments are given in bold, followed by our responses and modified text in our revised manuscript are given in quotes, italics and blue.

**Response to Reviewer 1:**

1. **In this study the UKCA chemistry-climate model is enhanced by incorporating reactive VOC tracers into the UKCA gas-phase chemistry scheme in order to better represent urban and regional-scale O$_3$ photochemistry, and then applied to quantify the differences in chemical environment for surface O$_3$ for six major industrial regions across China in summer 2016. This study is well organized and clearly written on an interesting topic – how to effectively control tropospheric ozone pollution in China. Here are some specific comments.**

We thank the reviewer for their positive comments here, and address specific concerns below.

2. **Six cities are chosen because they are located in the heavily populated regions with high emissions, but their climate is different, for example, Chongqing is often cloudy and foggy, and especially hot and muggy in summer. The amount of sunshine in Chongqing is poor, and the only months when the sunshine hours exceed 40% (but they still remain below 50%) are July and August; while Beijing has strange weather—extreme hot and cold temperatures, and high humidity to no humidity. It has glorious sunshine year-round and is hot and rainy in summer, especially in July and August. In the study impacts of meteorological conditions on local O$_3$ production rates are not discussed, and model results are not evaluated against observed meteorological parameters.**

We thank the reviewer for this comment and have investigated meteorological conditions further. We agree that meteorological conditions are important for O$_3$ formation, and many studies have focused on this aspect specifically (Gong and Liao, 2019; Liu and Wang, 2020; Shi et al., 2020). We have added additional text to introduce the importance of meteorology to O$_3$ formation in the introductory section of the paper:

Page 2, line 47:
*"Meteorological processes also affect O$_3$ formation through temperature, humidity, clouds, precipitation and biogenic emissions, and a number of papers have studied meteorological impacts on O$_3$ over China (Gong and Liao, 2019; Liu and Wang, 2020; Shi et al., 2020). However, emission controls are the primary strategies used to reduce*

*O₃ pollution and we focus on these for this study, as their effectiveness for different regions has not been fully investigated."*

We emphasise that the model is nudged to ECMWF ERA-interim meteorological reanalysis data, hence the meteorology over these broad regions is generally simulated well. We show a meteorological evaluation for temperature and humidity for the regions we consider in Table 1. Typically, high temperature and low humidity benefit $O_3$ production in China, as outlined in the studies listed above. Observed daily mean data have been obtained from the national meteorological data center http://data.cma.cn/.

**Table 1:** Comparison of daily mean of observed and simulated surface temperature (°C) and relative humidity (%) for six regions of China during June, July, August, 2016.

| | Temperature (°C) | | | | |
|---|---|---|---|---|---|
| | Obs. | Sim. | Bias | RMSE | r |
| Beijing | 24.0 | 21.7 | -2.3 | 2.7 | 0.8 |
| Shijiazhuang | 26.3 | 24.3 | -2.0 | 2.8 | 0.7 |
| Shanghai | 27.6 | 25.6 | -2.0 | 2.6 | 0.9 |
| Nanjing | 27.8 | 26.9 | -0.9 | 2.0 | 0.9 |
| Guangzhou | 28.3 | 27.3 | -1.0 | 1.8 | 0.5 |
| Chongqing | 29.4 | 27.5 | -1.9 | 3.0 | 0.8 |

| | Relative Humidity (%) | | | | |
|---|---|---|---|---|---|
| | Obs. | Sim. | Bias | RMSE | r |
| Beijing | 67.0 | 79.6 | 12.6 | 14.6 | 0.8 |
| Shijiazhuang | 72.1 | 80.8 | 8.7 | 13.2 | 0.8 |
| Shanghai | 80.4 | 79.8 | -0.6 | 5.8 | 0.8 |
| Nanjing | 77.3 | 77.9 | 0.6 | 5.5 | 0.9 |
| Guangzhou | 85.5 | 82.8 | -2.7 | 6.0 | 0.6 |
| Chongqing | 68.2 | 81.4 | 13.2 | 16.5 | 0.5 |

From Table 1, we can see that the model is biased low for simulated temperature, although this is partly due to use of temperature in the lowest model layer rather than at 2 m altitude. However, the model successfully captures regional differences in temperature – the highest in Chongqing and the lowest in BTH regions e.g. Beijing and Shijiazhuang. Relative humidity in Shanghai, Nanjing, Guangzhou is relatively well simulated; the highest relative humidity for Guangzhou is also captured by the model. The model biases for Beijing and Chongqing are larger, but this is likely to be due to their proximity to mountainous areas influencing the respective model grid boxes. Higher humidity promotes $O_3$ destruction hence may lead to lower surface $O_3$ levels, and this could partly explain negative model biases in daily mean surface $O_3$ levels for Beijing. However, we note that temperature and humidity are not the main influences on surface $O_3$ concentrations in Chongqing, and topography is a more important factor as discussed in the paper. Given our use of reliable ECMWF ERA-interim meteorological reanalysis data and the uncertainties associated with meteorological

evaluation, we choose not to include this discussion in the revised manuscript. For clarity we have added the following text to section 3:

Page 11, line 231:
*"Given our use of reliable meteorological reanalysis data, we note that meteorology is not the main influence on the model biases. We therefore investigate $O_3$ chemical environments in different regions to explore regional differences below."*

3. **Figure 3 shows that the model overestimates $O_3$ systematically in Chongqing, and observed low concentrations (<35 ppb) are not reproduced. As statistical indicators are limited and independent in this study, high correlation coefficient r does not indicate that the modeled concentrations in Chongqing are acceptable, because seasonal averaged diurnal variations in $O_3$ over high emission areas are quite easy to be simulated. Beside the explanation in lines 165-169, would the meteorological fields be attributable?**

Chongqing is not simulated as well as other regions as we note in the text. The overestimation of surface $O_3$ levels for Chongqing is likely linked to the complex topography as discussed in section 3. Surface $O_3$ in Chongqing is representative of higher surface altitudes characteristic of the region, leading to a systematic high bias compared with observations, and a corresponding low bias for $NO_2$ concentrations. Our results suggest that the region of Chongqing represents a $NO_x$ limited regime, hence the underestimated $NO_2$ levels is not the cause to the overpredicted $O_3$ levels. This is consistent with satellite-based assessments of ozone sensitivity in China for 2016 (Wang et al., 2021). It hence suggests that complex transport patterns associated with the topography and the higher altitude of the region are the most important factors influencing $O_3$ levels rather than meteorology as discussed above. We have added the following text to section 3:

Page 11, line 236:
*"This underestimation may lead to overestimated $O_3$ concentrations in a VOC-limited regime and underestimated $O_3$ in a $NO_x$-limited regime. While underestimated $NO_x$ concentrations may reflect underestimated $NO_x$ emissions, it is more likely to arise from the effects of dilution on $NO_x$."*

4. **From Figure 4, we can also find that $NO_2$ in Chongqing is not reproduced well, and daytime $NO_2$ is underestimated at all sites. How would the underestimation of daytime $NO_2$ impact on $O_3$ production sensitivity?**

We agree that the underestimation of daytime $NO_2$ concentrations will influence $O_3$ sensitivity. However, higher $NO_x$ levels lead to a more VOC limited regime so all regions except Chongqing would still be in VOC limited regimes if $NO_x$ levels were increased. From the $O_3$ isopleth plot in Fig 8, we can see that increasing $NO_x$ levels would leave Chongqing in the transition regime or still in a $NO_x$ limited regime.

However, to address this concern we have toned down our statement that 'Chongqing is $NO_x$ limited' to accommodate the local underestimation of $NO_2$ concentrations. Text added in section 6 and 7 in the revised manuscript for clarity are as follows:

Page 17, line 348:
*"We note that our conclusion of $NO_x$ limitation in Chongqing may be sensitive to our underestimation of $NO_2$ levels (section 3), and that with higher $NO_2$ Chongqing may shift from the $NO_x$ limited regime to the VOC limited regime. However, satellite observation based studies have also identified this region as one that is largely $NO_x$ limited, in contrast to the heavily populated coastal regions (Wang et al., 2021)."*

Page 19, line 397:
*"From the $O_3$ isopleth for Chongqing, we can see that even if $NO_x$ emissions were increased by 40 %, Chongqing would still be in $NO_x$ limited or transition regimes. This suggests that Chongqing is still far away from a VOC limited regime."*

5. **Bye the way, $O_3$ and $NO_2$ from the surface monitoring networks of China are usually recorded in the unit of μg/m3, how are they converted to ppb?**

The units of μg/m$^3$ are converted to ppb based on temperature and pressure as follows:

μg/m$^3$ = ppb * molecular weight/molecular volume (litres)

where molecular volume = 22.41 * (temperature/273.15) * (1013/pressure)
T = absolute temperature (K)
P = atmospheric pressure (hPa)
Chemical species in units of mixing ratio (kg/kg) are output by the model and then converted to ppb. This unit conversion is a standard procedure and so we do not include a description of the approach in the paper.

6. **Tropospheric $O_3$ concentrations are functions of the chain lengths of $NO_x$ and Hox radical catalytic cycles, and ozone production rates depend on not only $NO_x$ and VOCs concentrations, but also actinic flux and temperature. The conclusion of $O_3$ production across all regions except Chongqing being VOC limited needs to studies for different weather conditions.**

$NO_x$ and VOCs are the main primary $O_3$ precursor species emitted from emission sources. Therefore, we construct $O_3$ response surfaces with respect to changing $NO_x$ and VOC emissions, and our aim is to investigate the impacts of changing $NO_x$ and VOC emissions on surface $O_3$ concentrations. We choose to represent $O_3$ sensitivity to $NO_x$ and VOC emissions because it is largely driven by ambient abundances of $NO_x$ and VOCs – reflected by VOC and $NO_x$ limited regimes. We agree that actinic flux and temperature alter $O_3$ production rates through their effects on photolysis rates and chemical reaction rates, and these are fully captured in our model chemistry scheme, which includes 101 species and 348 reactions (see section 2.1). We note that $O_3$

sensitivity is affected by photolysis rates and temperature, but highlight that we use the meteorological conditions appropriate to each region to calculate the sensitivity. It is a good point that $O_3$ sensitivity could be investigated in future studies for different weather conditions, and it is also worthy investigating how $O_3$ sensitivity changes throughout a day. However, this study aims to provide a broad assessment of regional $O_3$ sensitivity with the implication of emission control strategies in the summer time, and focus on specific time periods or unusual meteorological conditions is not necessary for this.

**7. Generally model simulated $O_3$ production rates are quite sensitive to the chemical mechanism beside other model inputs. As stated in lines 101-105, VOCs such as alkenes and aromatics are abundant in industrial areas of China, but why is ethene not incorporated into the UKCA gas-phase chemistry scheme?**

The extended chemistry scheme for UKCA introduced in this study provides highly reactive VOC species to permit representation of more active photochemical environments. We incorporate $C_3H_6$, $C_4H_{10}$ and toluene as proxies for the principal chemically reactive VOC families: alkenes, alkanes and aromatics. The tracer $C_3H_6$ thus implicitly includes ethene as a part of the alkene family. This approach to VOC chemistry is described in detail in Stockwell et al. (1990). We highlight that this extended chemistry scheme permits a more realistic VOC oxidation environment, and is more suitable for high emission areas than the original chemistry scheme that did not include these short-lived VOC tracers.

**Response to Reviewer 2:**

1. **This manuscript investigated chemical environment for surface $O_3$ for six major industrial regions across China in summer 2016. Detailed chemistry-climate model simulations were employed to diagnose ozone sensitivity to precursors and contrast the effectiveness of different measures to reduce surface $O_3$ concentrations. This manuscript is helpful to understand ozone pollution mechanism in Chinese cities, and within the scope of ACP. I think it is publishable in ACP after my following concerns are addressed.**

We thank the reviewer for their positive comments, and specific concerns are addressed below.

2. **Line 215: The gross rate of production $P(O_3)$ actually represents the production rate of $O_X$ ($O_3$ + $NO_2$) through the reaction $HO_2$ ($RO_2$) +NO. Therefore, the net ozone production rate should include the loss term $NO_2$+OH (Wang et al., 2019. doi.org/10.5194/acp-19-9413-2019). In addition to OH+$NO_2$ and $RO_2$+$NO_2$, the loss of NOx should also include $RO_2$+NO and OH+HONO When calculating OPE. Please give specific quantification even though these reactions play a minor role in the loss of NOx.**

We thank the reviewer for this point. The calculation of $O_3$ production that the reviewer points out is based on including $NO_2$ in the definition of odd oxygen ($O_x$). We note that there are different definitions of $O_x$ and sometimes $O_3$ and $NO_2$ are together defined as $O_x$. These wider definitions include $O_3$, $O(^1D)$, $O(^3P)$, $NO_2$ and various other $NO_y$ species ($NO_3$, $N_2O_5$, $HO_2NO_2$, $HNO_3$, PAN, MPAN), see Section 4 in Horowitz et al. (2003). There is still ongoing debate about the best way to consider $O_x$, and recent studies have used an even wider definition (Bates and Jacob, 2020). However, we choose to take a simpler and more traditional definition of $O_x$ including only $O_3$ and O species, following Kleinman et al. (1997). No approach provides a complete assessment of all odd oxygen, but we feel strongly that the simplicity and greater interpretability of this more traditional approach outweighs the benefits of including a large number of minor species, while also allowing direct comparability with earlier studies which have used the same approach. Using this approach, $O_3$ production is the sum of $HO_2/RO_2$ + NO reactions and $O_3$ loss is the sum of direct reactions of $O_3$ with OH, $HO_2$ and alkenes, and $O(^1D)$ + $H_2O$. We now state this clearly in section 5:

Page 15, line 298:
*"We define the net $O_3$ production rate (ppb/h) as the gross rate of production of $O_3$, $P(O_3)$, from the reactions $HO_2$ + NO and $RO_2$ + NO minus the gross rate of loss of $O_3$, $L(O_3)$, from the reactions $O(^1D)$ + $H_2O$, $O_3$ + OH, $O_3$ + $HO_2$ and $O_3$ + VOCs."*

We regard to the reaction $NO_2$ + OH as the principal loss of $NO_x$ but it is not a loss of $O_3$ because we do not include $NO_2$ as part of odd oxygen – this is an advantage of the

simpler definition of $O_x$. The $NO_x$ loss term is defined as $NO_2 + OH$ in Liu et al. (1987), assuming that the lifetime of $NO_x$ is determined by the formation and deposition of $HNO_3$ in the daytime. However, the formation of organic nitrate ($RONO_2$) is also a minor loss of $NO_2$ as the reviewer points out. We therefore include this reaction in the $NO_x$ loss term as the reviewer suggests. It also has minor effects on the calculation of OPE. To account for the reviewer's suggestion, we have redefined the $NO_x$ loss and OPE in the paper and have updated Fig. 6. Our results in the text are unchanged. We have also introduced $RONO_2$ in the introductory section. Oxidation of HONO by OH does not remove $NO_x$ so this reaction is not included in the $NO_x$ loss term. The modified text is shown below.

Page 2, Line 60:
*"However, at high $NO_x$ concentrations, nitric acid ($HNO_3$), peroxy nitrates ($RO_2NO_2$) and organic nitrates ($RONO_2$) are easily formed as $NO_x$ reacts with OH and $RO_2$. These species are the main sinks of radicals and $NO_x$, and are readily removed from the atmosphere by deposition or exported to remote areas (Horowitz et al., 1998). Therefore, increasing $NO_x$ concentrations increase $O_3$ production, but also accelerate the formation of $NO_x$ sinks, leading to less efficient $O_3$ formation."*

Page 15, Line 308:
*"The loss of $NO_x$, $L(NO_x)$, is principally determined by the reactions $OH + NO_2$, $RO_2 + NO_2$ and $RO + NO_2$, which produce $HNO_3$, $RO_2NO_2$ and $RONO_2$ respectively."*

[Figure]

**Figure 6:** Simulated surface daytime **(a)** net $O_3$ production rates, gross $O_3$ production rates and gross $O_3$ loss rates (ppb/h) **(b)** gross $O_3$ production rates and $NO_x$ loss rates (ppb/h) **(c)** OPE (unitless) for the six industrial regions in JJA, 2016, China.

3. **Figure 4 shows significant underestimation for $NO_2$ in daytime, but overestimation for $NO_2$ at nighttime. The overestimation of $NO_2$ at night maybe related to underestimated nighttime chemistry such as the removal of $NO_3$ and $N_2O_5$ through heterogenous uptake (Li et al., 2018; Li et al., 2019). A short discuss should be performed. Additionally, how do these underestimation and overestimation for NO2 influence your diagnosis of ozone sensitivity? For example, the underestimation of NO2 in Chongqing will lead to more NOx-**

**limited, which likely misleads the actual situation.**

The heterogeneous chemistry scheme in UKCA includes heterogenous uptake for $NO_3$ and $N_2O_5$ – the conversion of $N_2O_5$ to aqueous $HNO_3$. Uptake coefficients are estimated for the different aerosol types included in the GLOMAP aerosol scheme. In UKCA, the lack of nitrate aerosol in the aerosol scheme may result in the lower uptake of nitrogen (Archibald et al., 2020), particularly in regions with high $NO_x$ emissions. Therefore, the heterogeneous removal of nitrogen may be biased, potentially leading to higher $NO_2$ and lower $O_3$ concentrations at nighttime. We have now added further text to discuss the potential impacts of heterogeneous processes in section 3 on model evaluation:

Page 7, Line 202:
*"In addition, nighttime heterogeneous uptake of nitrogen on aerosols remains highly uncertain due to the complexity in estimating uptake coefficients for different aerosol composition/mixing states (Lowe et al., 2015; Tham et al., 2018). In UKCA, the lack of nitrate aerosol in the aerosol scheme may result in a lower uptake of nitrogen (Archibald et al., 2020), particularly in regions with high $NO_x$ emissions. Therefore, there may be a bias in the heterogeneous removal of nitrogen, potentially leading to higher $NO_2$ and lower $O_3$ concentrations at nighttime."*

However, we note that we focus on daytime $O_3$ levels and $O_3$ sensitivity, and therefore uncertainties in nighttime chemistry have a relatively small influence on our results.

As in our response to the reviewer 1, we now include discussion of the impacts of underestimated $NO_2$ levels on $O_3$ sensitivity. We note that higher $NO_2$ levels for Beijing, Shijiazhuang, Shanghai, Nanjing, Guangzhou would lead to more VOC limited chemical environments, so this would not affect our conclusions. However, it is possible that higher $NO_x$ emissions might shift Chongqing into a transition regime (see Fig 8 in section 6). We have adjusted the statement that 'Chongqing is $NO_x$ limited' in sections 6 and 7 as follows:

Page 17, line 348:
*"We note that our conclusion of $NO_x$ limitation in Chongqing may be sensitive to our underestimation of $NO_2$ levels (section 3), and that with higher $NO_2$ Chongqing may shift from the $NO_x$ limited regime to the VOC limited regime. However, satellite observation based studies have also identified this region as one that is largely $NO_x$ limited, in contrast to the heavily populated coastal regions (Wang et al., 2021)."*

Page 19, line 397:
*"From the $O_3$ isopleth for Chongqing, we can see that even if $NO_x$ emissions were increased by 40 %, Chongqing would still be in $NO_x$ limited or transition regimes. This suggests that Chongqing is still far away from a VOC limited regime."*

4. **Figure 8. shows ozone increased from 70 ppb to over 80 ppb during 2013-2019. However, observed ozone concentrations in Beijing didn't increased significantly during the period or decreased after 2015 in spite that ozone increased over North China Plain (Lu et al., 2018. DOI: 10.1021/acs.estlett.8b00366; Tang et al., 2020. doi.org/10.1016/j.atmosres.2020.105333). This needs further explanations.**

Tang et al. (2021) suggest that the chemical environment for Beijing is now $NO_x$ limited hence reductions of $NO_x$ emissions lower surface $O_3$ levels. However, our study focuses on the regional scale situation rather than on urban measurement locations. Many studies have shown an overall increasing trend in $O_3$ levels on the North China Plain in recent years, as the reviewer notes. It would not be representative to use a single measurement site in the outskirts of the city to interpret the overall trend of the whole region. We have added text in the conclusion section to emphasise the aim of this study:

Page 23, line 476:
*"This study hence provides a broad assessment of the $O_3$ sensitivities for these regions with the implication of emission control strategies."*

5. **Line 270: How do you obtain VOC and NOx emissions in 2018 and 2019 given that Cheng et al (2019) just estimated emissions during 2013-2017. Please give specific description.**

We assume that $NO_x$ and VOC emissions in the Beijing region continue to follow the same trend as they do between 2013 and 2016. We have now added further text to clarify this in section 7. We already note in the conclusions that emission inventory development is a limitation for accurately predicting $O_3$ trends.

Page 18, line 383:
*"For context, Fig. 8a also shows the simulated daytime $O_3$ changes between 2013 and 2019 in the Beijing region assuming that the emission changes observed between 2013 and 2016 continue at the same rate until 2019 (Cheng et al., 2019)."*

6. **Line 145: There are only 450 measurement stations in 2013, growing to 1,500 stations in 2017 and 1670 stations in 2019.**

Yes, the number of measurement sites in China increased substantially after 2013. The text in the paper hase been modified to reflect this.

Page 6, Line 177:
*"450 measurement stations in China started operating in 2013, growing rapidly to 1670 stations by 2019."*

7. **Line 300: "summer-mean ozone" should be "daily mean ozone".**

We replace 'summer-mean ozone' with 'summer daily-mean ozone' throughout the text in section 7.

Below are all references used in the responses. Added references in the revised manuscript are shown as track changes.

**References:**

Archibald, A. T., O'Connor, F. M., Abraham, N. L., Archer-Nicholls, S., Chipperfield, M. P., Dalvi, M., Folberth, G. A., Dennison, F., Dhomse, S. S., Griffiths, P. T., Hardacre, C., Hewitt, A. J., Hill, R. S., Johnson, C. E., Keeble, J., Kohler, M. O., Morgenstern, O., Mulcahy, J. P., Ordonez, C., Pope, R. J., Rumbold, S. T., Russo, M. R., Savage, N. H., Sellar, A., Stringer, M., Turnock, S. T., Wild, O., and Zeng, G.: Description and evaluation of the UKCA stratosphere-troposphere chemistry scheme (StratTrop vn 1.0) implemented in UKESM1, Geoscientific Model Development, 13, 1223-1266, 10.5194/gmd-13-1223-2020, 2020.

Bates, K. H., and Jacob, D. J.: An Expanded Definition of the Odd Oxygen Family for Tropospheric Ozone Budgets: Implications for Ozone Lifetime and Stratospheric Influence, Geophysical Research Letters, 47, 10.1029/2019gl084486, 2020.

Cheng, J., Su, J. P., Cui, T., Li, X., Dong, X., Sun, F., Yang, Y. Y., Tong, D., Zheng, Y. X., Li, Y. S., Li, J. X., Zhang, Q., and He, K. B.: Dominant role of emission reduction in PM2.5 air quality improvement in Beijing during 2013-2017: a model-based decomposition analysis, Atmospheric Chemistry and Physics, 19, 6125-6146, 10.5194/acp-19-6125-2019, 2019.

Gong, C., and Liao, H.: A typical weather pattern for ozone pollution events in North China, Atmos. Chem. Phys., 19, 13725-13740, 10.5194/acp-19-13725-2019, 2019.

Horowitz, L. W., Liang, J. Y., Gardner, G. M., and Jacob, D. J.: Export of reactive nitrogen from North America during summertime: Sensitivity to hydrocarbon chemistry, Journal of Geophysical Research-Atmospheres, 103, 13451-13476, 10.1029/97jd03142, 1998.

Horowitz, L. W., Walters, S., Mauzerall, D. L., Emmons, L. K., Rasch, P. J., Granier, C., Tie, X. X., Lamarque, J. F., Schultz, M. G., Tyndall, G. S., Orlando, J. J., and Brasseur, G. P.: A global simulation of tropospheric ozone and related tracers: Description and evaluation of MOZART, version 2, Journal of Geophysical Research-Atmospheres, 108, 10.1029/2002jd002853, 2003.

Kleinman, L. I., Daum, P. H., Lee, J. H., Lee, Y. N., Nunnermacker, L. J., Springston, S. R., Newman, L., WeinsteinLloyd, J., and Sillman, S.: Dependence of ozone production on NO and hydrocarbons in the troposphere, Geophysical Research Letters, 24, 2299-2302, 10.1029/97gl02279, 1997.

Liu, S. C., Trainer, M., Fehsenfeld, F. C., Parrish, D. D., Williams, E. J., Fahey, D. W., Hubler, G., and Murphy, P. C.: Ozone production in the rural troposphere and the implications for regional and global ozone distributions, Journal of Geophysical Research-Atmospheres, 92, 4191-4207, 10.1029/JD092iD04p04191, 1987.

Liu, Y., and Wang, T.: Worsening urban ozone pollution in China from 2013 to 2017 –

Part 1: The complex and varying roles of meteorology, Atmos. Chem. Phys., 20, 6305-6321, 10.5194/acp-20-6305-2020, 2020.

Lowe, D., Archer-Nicholls, S., Morgan, W., Allan, J., Utembe, S., Ouyang, B., Aruffo, E., Le Breton, M., Zaveri, R. A., Di Carlo, P., Percival, C., Coe, H., Jones, R., and McFiggans, G.: WRF-Chem model predictions of the regional impacts of N2O5 heterogeneous processes on night-time chemistry over north-western Europe, Atmospheric Chemistry and Physics, 15, 1385-1409, 10.5194/acp-15-1385-2015, 2015.

Shi, Z., Huang, L., Li, J., Ying, Q., Zhang, H., and Hu, J.: Sensitivity analysis of the surface ozone and fine particulate matter to meteorological parameters in China, Atmos. Chem. Phys., 20, 13455-13466, 10.5194/acp-20-13455-2020, 2020.

Stockwell, W. R., Middleton, P., Chang, J. S., and Tang, X. J. J. o. G. R. A.: The second generation regional acid deposition model chemical mechanism for regional air quality modeling, 95, 16343-16367, 1990.

Tang, G. Q., Liu, Y. S., Zhang, J. Q., Liu, B. X., Li, Q. H., Sun, J., Wang, Y. H., Xuan, Y. J., Li, Y. T., Pan, J. X., Li, X., and Wang, Y. S.: Bypassing the NOx titration trap in ozone pollution control in Beijing, Atmospheric Research, 249, 10.1016/j.atmosres.2020.105333, 2021.

Tham, Y. J., Wang, Z., Li, Q. Y., Wang, W. H., Wang, X. F., Lu, K. D., Ma, N., Yan, C., Kecorius, S., Wiedensohler, A., Zhang, Y. H., and Wang, T.: Heterogeneous N2O5 uptake coefficient and production yield of ClNO2 in polluted northern China: roles of aerosol water content and chemical composition, Atmospheric Chemistry and Physics, 18, 13155-13171, 10.5194/acp-18-13155-2018, 2018.

Wang, W., van der A, R., Ding, J., van Weele, M., and Cheng, T.: Spatial and temporal changes of the ozone sensitivity in China based on satellite and ground-based observations, Atmos. Chem. Phys., 21, 7253-7269, 10.5194/acp-21-7253-2021, 2021.

---

## Author Response (AR2)

Dear editor:

We thank you for your suggestions for further improvements to our manuscript. We have taken your concerns into account and give responses below on behalf of all authors. For clarity, editor comments are given in bold, followed by our responses. Modified text in our revised manuscript are given in quotes, italics and blue.

**Responses:**
1. **This time the co-editor has evaluated the revised manuscript, on behalf of the two reviewers. I find the revised manuscript has been improved, based on the reviewers' comments. However, the following minor points still need to be clarified and justified. I would appreciate it if the authors could take them into account.**

**(line numbers are those for acp-2020-1251-ATC1.pdf, with track changes)**

We thank the editor for the recognition of our work here, and address specific concerns below.

2. **Have the authors compared the simulated VOC concentration levels (Fig. 5c and d) with the observations to confirm the adequacy? For biogenic VOC levels over 16 ppb (line 275), any support observations are present? The concentration range would be too high for isoprene/monoterpenes - is it mostly contributed from less reactive species such as methanol? Some discussion should be added, as this point is also critical in the determination of the regimes.**

Since VOC concentrations are not measured by the standard air pollution monitoring networks in China, we cannot provide a comprehensive evaluation. Methanol is included in biogenic VOCs in Fig. 5, and contributes to the overall biogenic VOC concentrations. The editor is correct to note that methanol makes a substantial contribution to biogenic VOC over southern central China, as shown in Figure 1 below. Despite the relatively high concentrations of methanol, its reactivity is lower than most anthropogenic VOC species, and it makes relatively little contribution to $O_3$ sensitivity. In this study we have focused on the effect of anthropogenic VOC emissions, and we note that biogenic VOC are relatively low in all but one of the regions we investigate here. We now state this clearly in section 4 and 6:

Page 13, Line 272:
*"The distribution of biogenic VOC concentrations (including isoprene and methanol) differs from that of anthropogenic VOCs (Fig. 5c, 5d)."*

Page 17, Line 349:
*"The determination of $O_3$ sensitivity regimes here is based on the $O_3$ responses to decreasing anthropogenic $NO_x$ and/or VOC emissions, and any potential impacts of*

*changing BVOC emissions has not been assessed. Decreasing BVOC emissions may offset the increase in O₃ levels due to decreased NOₓ emissions for the NCP, the YRD and the PRD, and would make all regions more VOC limited. We note that our conclusion of NOₓ limitation in Chongqing may be sensitive to our underestimation of NO₂ levels (section 3), and to the higher BVOC emissions in this region, both of which reduce the ratio of NOₓ to VOC in the region (Table 3).*"

[Figure]

**Figure 1.** Spatial distributions of simulated surface daytime concentrations of isoprene and monoterpene (a) and methanol (b) in JJA, 2016, China.

**3. Line 257, Figure 5 and Table 3. How did the authors define the "daytime"? Any specific time period with the local time?**

In this study, we refer to MDA8 O₃ concentrations as daytime O₃ concentrations to enhance readability. Following the standard definition of MDA8, we use the Maximum Daily Average 8 hour concentrations calculated from consecutive 8 hour running mean values over 24 hours. We have then used the same time period (typically 11:00-18:00 or 12:00-19:00) for all species. We state this on line 255, but we have rephrased this to make it clearer.

Page 12, Line 255:
*"We use the standard definition of the Maximum Daily Average 8-hour (MDA8) Ozone metric, and consider this same time period for other species, which we refer to hereafter as daytime concentrations."*

**4. Section 6. How much were the total VOC CONCENTRATIONS (or reactivity) reduced with the 20% emission change? I am wondering if biogenic VOCs might have been dominant and thus there is no virtual change in the total concentrations of VOCs and their role, even if the emissions of "anthropogenic" VOCs are varied over a wide range (in Chongqing for example).**

This is an interesting and valuable point, and the editor is correct to point out that the higher BVOC emissions influence the result and partly explain the higher $NO_x$ needed to drive the region into VOC limitation. Our results reflect the effects of anthropogenic emission changes, which remain fully valid, but the differing underlying response may indeed reflect higher BVOC. In Table 3 we have shown the concentration of anthropogenic and biogenic VOC along with the ratio of $NO_x$ to total VOC. This highlights the different chemical environment in Chongqing, and provides an additional reason for the different $O_3$ sensitivity. We have now amended the text to acknowledge the contribution of the higher background BVOC in Chongqing:

Page 17, Line 352:
*"We note that our conclusion of $NO_x$ limitation in Chongqing may be sensitive to our underestimation of $NO_2$ levels (section 3), and to the higher BVOC emissions in this region, both of which reduce the ratio of $NO_x$ to VOC in the region (Table 3)."*

5. **Line 351. Satellite observations of NO2 and HCHO have large uncertainties (of several tens of percents, e.g., Pinardi et al., 2020 for OMI NO2) and therefore the determination of regimes from the ratio would not be very certain.**

**Pinardi, G., Van Roozendael, M., Hendrick, F., Theys, N., Abuhassan, N., Bais, A., Boersma, F., Cede, A., Chong, J., Donner, S., Drosoglou, T., Dzhola, A., Eskes, H., Fries, U., Granville, J., Herman, J. R., Holla, R., Hovila, J., Irie, H., Kanaya, Y., Karagkiozidis, D., Kouremeti, N., Lambert, J.-C., Ma, J., Peters, E., Piters, A., Postylyakov, O., Richter, A., Remmers, J., Takashima, H., Tiefengraber, M., Valks, P., Vlemmix, T., Wagner, T., and Wittrock, F.: Validation of tropospheric NO2 column measurements of GOME-2A and OMI using MAX-DOAS and direct sun network observations, Atmos. Meas. Tech., 13, 6141?6174, https://doi.org/10.5194/amt-13-6141-2020, 2020.**

We agree that estimating $O_3$ sensitivity from satellite data carries substantial uncertainty as the editor mentions but we note that it provides a very useful indication of $O_3$ sensitivity regimes. We acknowledge this and we now use the word "*suggest*" to replace "*identify*" in the manuscript. We have modified text in section 6:

Page 17, Line 354:
*"However, satellite observation based studies have also suggested this region as one that is largely $NO_x$ limited in 2016, in contrast to the heavily populated coastal regions (Wang et al., 2021)."*

6. **Figure 8f, lines 397-400. To my eye, the point with 40% NOx increase and 0% VOC change would be within the VOC limited side. In Figure 9a also, at the point of NOx emission change of 40%, surface O3 change is almost saturated. Please double check.**

The transition point for Chongqing lies at roughly 40% NO$_x$ emissions increase, as the editor notes. We hence remove sentences on lines 401 – 403 to avoid the confusion.

**7. Line 433. Although the authors simply state that all selected regions across the globe outside of China are NOx limited, I would suggest that the chosen resolution would affect and the real situation is not that simple. For example in the central Tokyo, when studied at a resolution of ca. 10 km, summertime ozone formation is clearly VOC limited (Inoue et al., 2019). The effectiveness of legislated VOC emissions reduction is seriously studied. I hope the authors could acknowledge this point. Maybe in line 433, after "NOx limited," it would be better to add "with the studied horizontal resolution".**

**Inoue, K., Tonokura, K. & Yamada, H. Modeling study on the spatial variation of the sensitivity of photochemical ozone concentrations and population exposure to VOC emission reductions in Japan. Air Qual Atmos Health 12, 1035?1047 (2019). https://doi.org/10.1007/s11869-019-00720-w**

We agree that these conclusions are scale-dependent, and note that our results apply to the model scales resolved here that are representative of wider urban regions. Clearly smaller regions with more intense emissions may still be VOC limited, but these lie within wider regions that are NO$_x$ limited at the scales considered here. We have added the following text in the manuscript:

Page 20, Line 437:
*"We find that all selected high emission regions across the globe outside of China are NO$_x$ limited at the model resolution considered here, such that NO$_x$ emissions decreases yield regional O$_3$ decreases. Current levels of NO$_x$ emissions in these regions are considerably lower than for the industrial regions of China, reflecting the different O$_3$ sensitivity regimes (Table 5). We note that these results apply to the wide urban regions considered here, and that local O$_3$ sensitivity in some parts of these regions may be different."*